# Research

ecology, palaeontology

spatial bias, latitudinal diversity gradient, macroecology, global diversity, marine invertebrates

**Author for correspondence:**
Lewis A. Jones
e-mail: lewisa.jones@outlook.com

# Spatial sampling heterogeneity limits the detectability of deep time latitudinal biodiversity gradients

Lewis A. Jones[1], Christopher D. Dean[2], Philip D. Mannion[3],
Alexander Farnsworth[4] and Peter A. Allison[1]

[1]Department of Earth Science and Engineering, Imperial College London, South Kensington,
London SW7 2AZ, UK
[2]School of Geography, Earth and Environmental Sciences, University of Birmingham, Edgbaston,
Birmingham B15 2TT, UK
[3]Department of Earth Sciences, University College London, London WC1E 6BT, UK
[4]School of Geographical Sciences, University of Bristol, Bristol BS8 1TH, UK

LAJ, 0000-0003-3902-8986; CDD, 0000-0001-6471-6903; PDM, 0000-0002-9361-6941

The latitudinal biodiversity gradient (LBG), in which species richness decreases from tropical to polar regions, is a pervasive pattern of the modern biosphere. Although the distribution of fossil occurrences suggests this pattern has varied through deep time, the recognition of palaeobiogeographic patterns is hampered by geological and anthropogenic biases. In particular, spatial sampling heterogeneity has the capacity to impact upon the reconstruction of deep time LBGs. Here we use a simulation framework to test the detectability of three different types of LBG (flat, unimodal and bimodal) over the last 300 Myr. We show that heterogeneity in spatial sampling significantly impacts upon the detectability of genuine LBGs, with known biodiversity patterns regularly obscured after applying the spatial sampling window of fossil collections. Sampling-standardization aids the reconstruction of relative biodiversity gradients, but cannot account for artefactual absences introduced by geological and anthropogenic biases. Therefore, we argue that some previous studies might have failed to recover the 'true' LBG type owing to incomplete and heterogeneous sampling, particularly between 200 and 20 Ma. Furthermore, these issues also have the potential to bias global estimates of past biodiversity, as well as inhibit the recognition of extinction and radiation events.

## 1. Introduction

The fossil record is a spatial and temporal archive of past biodiversity that facilitates the evaluation of trends in evolution, extinction and biological recovery through deep time. It also provides critical context for understanding the impact of ongoing and future climatic change on global ecosystems, and offers potentially vital information for current conservation efforts [1,2]. However, this archive is impacted by preservational and sampling biases that render it incomplete and potentially misleading [3–9]. Despite a growing number of mitigating techniques (e.g. [10]), understanding the degree to which these biases control our perception of spatial biodiversity patterns in deep time is still particularly challenging [9,11]. What questions can palaeobiologists truly ask of the fossil record given the quality of data? Is it possible to evaluate macroecological patterns throughout Earth's history (e.g. [12–14]), or is there a limit to the scale of the questions that can be asked of a biased dataset?

The latitudinal biodiversity gradient (LBG), in which species richness decreases from tropical to polar latitudes, is one of the most recognizable global macroecological patterns today. Although the present-day LBG has

been extensively documented and its causes debated for over two centuries [15–17], it is only in recent decades that large-scale variation in biodiversity has been considered within a deep time context [18–21]. Fossil occurrence data suggest that the 'modern-type' (unimodal) LBG was not always present, with a range of taxonomic groups across different environments characterized by flattened or even bimodal gradients in deep time (e.g. [13,14,22–30]). However, the extent to which these latitudinal patterns are the result of inherent biases (such as variable fossil preservation, stratigraphic completeness and geographical sampling bias), as opposed to biological processes, remains uncertain (e.g. [3–9]).

Several methodologies have been developed to standardize sampling and mitigate problems associated with uneven 'raw' occurrence data [31]. These approaches have most commonly been applied to allow comparisons between temporal bins (e.g. [9,10,31–34]). However, it is also recognized that spatial bias in sampling (e.g. preferential sampling in western Europe and North America) also skews observed palaeobiogeographic patterns [3–9,11,35,36]. Although sampling-standardization approaches have been implemented between palaeolatitudinal bins in some studies (e.g. [13,14]), these methods do not explicitly address the spatio-temporal heterogeneity of sampled geographical area [9,11,37]. As inherently spatial patterns, estimations of LBGs in deep time have the capacity to be particularly impacted by this bias: variation in the geographical area sampled by palaeontologists, between different palaeolatitudinal bands, can result in radically different sizes of sampled occurrences, which is in turn a strong control on taxon counts [11,37]. Consequently, despite the critical information contained within the fossil record, it is possible that limited and heterogeneous spatial sampling has prevented the reconstruction of genuine LBGs in deep time.

Here, we use a simulation framework to test the impact of spatial sampling bias on deep time palaeobiogeographic patterns in shallow marine environments. We simulated occurrence data conforming to three different types of LBG (flat, unimodal and bimodal) using palaeogeographies of 56 stratigraphic stages spanning the beginning of the Permian, to the end of the Neogene (Asselian–Piacenzian; 298.9–2.58 Ma). This time period covers a range of major Earth system changes, including transitions from greenhouse to icehouse climatic regimes, which are considered to be a major driver in the spatial distribution of biodiversity [20]. The simulated data were then spatially filtered using the global palaeogeographic sampling window of fossil collections for each stage (sourced from the Paleobiology Database (PBDB)), retaining only simulated occurrences that were found within $1° \times 1°$ grid cells that contained a real fossil collection. Simulated (using the raw dataset) and sampled (using the filtered dataset) palaeolatitudinal biodiversity curves were then constructed and compared to quantify the data lost between the 'known' and sampled fossil record. We also evaluated the use of sampling-standardization methods in reconstructing different types of LBGs in the light of variable spatial sampling through time. Finally, we document the palaeogeographic evolution of spatial sampling through time, and its impact on our understanding of the LBG in deep time. Our approach emulates how the true distribution of biodiversity might be masked by the 'known' spatial sampling window, and tests whether genuine LBGs can be reconstructed from the fossil record (figure 1).

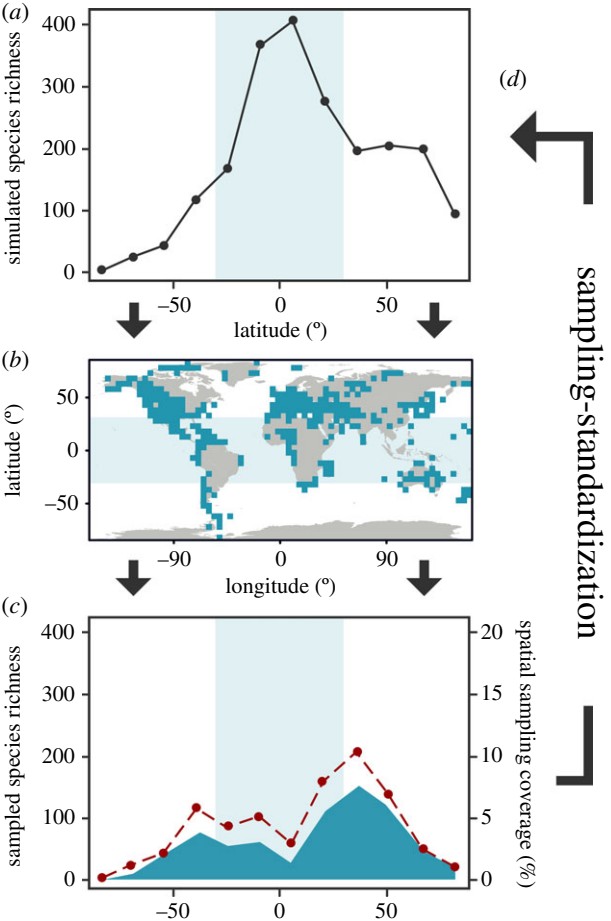

**Figure 1.** Schematic diagram of the workflow of this study. (*a*) Species' occurrences and diversity are simulated to conform to a 'known' latitudinal biodiversity gradient; (*b*) the simulated species' occurrences are spatially sampled by the 'known' sampling window (i.e. fossil collections); (*c*) the latitudinal biodiversity gradient is computed from the sampled simulated occurrences to assess how representative it is of the simulated 'known' latitudinal biodiversity gradient; (*d*) classical rarefaction is applied to assess whether sampling-standardization can aid reconstruction of relative diversity patterns. (*a*–*c*) Light blue shaded area represents the extent of the tropics. (*b*) Blue cells indicate sampled localities. (*c*) Dark blue ribbon depicts the latitudinal spatial sampling coverage (corresponding to blue cells in *b*). (Online version in colour.)

## 2. Methods

### (a) Simulation overview

We simulated marine invertebrate occurrence data constrained to three different types of LBG for the last 300 Myr: (i) an LBG with no specified peak in biodiversity (flat-type); (ii) a tropical peak in biodiversity with a sharp decline towards the poles (unimodal-type); and (iii) temperate peaks in biodiversity that decline towards the poles and equator (bimodal-type) (electronic supplementary material, figures S1–S2). All simulations were performed in R ver. 3.6.2 [38] and the code is available on GitHub (https://github.com/LewisAJones/LBG_sim). Below, we provide an overview of the simulation protocol (see the electronic supplementary material, figure S1 for a graphical representation).

To generate realistic LBGs, we simulated (with 100 replications) the distribution of 1000 species on stage-level (Asselian (Permian) to Piacenzian (Neogene)) palaeogeographies ($1° \times 1°$) using weighted probability grids equating to each LBG type (electronic supplementary material, figure S3). Probability grids were used to define an initial point for each 'virtual' species

(e.g. in a unimodal-type LBG, this point would preferentially be allocated in the tropics). This point was then used to define the centre of a distance probability grid on which the occurrences of each species were generated (electronic supplementary material, figure S1). We opted to simulate a group constrained to shallow marine environments (less than or equal to 200 m depth, approximating the photic zone), and therefore clipped probability grids prior to initial point generation using digital elevation models (DEMs) provided by Getech PLC. These DEMs provide global gridded data ($1° \times 1°$) on topography and bathymetry at stratigraphic stage level [39,40]. In order to produce realistic distributions of species, we assessed empirical range size and occurrence frequency distributions of present-day species within five major marine invertebrate groups (anthozoans, bivalves, brachiopods, echinoids and sponges), from data housed within the Ocean Biogeographic Information System (https://obis.org/) (electronic supplementary material, figures S4–S6). Using these data, we randomly computed the extent (range size; electronic supplementary material, table S1) of the distance probability grid and the number of occurrences for each species (between 1 and 300), based on an exponential decay curve (electronic supplementary material, figures S4–S6). We used Kolmogorov–Smirnov two-sample tests to assess the similarity of our simulated species' geographical range sizes and occurrence frequency distributions to those of empirical groups (anthozoans, bivalves, brachiopods, echinoids and sponges). These tests indicate a mixture of statistically significant ($p < 0.05$) and insignificant ($p \geq 0.05$) differences depending on the pairwise combination of simulated and empirical groups, and the metric used to compare them (electronic supplementary material, tables S2–S3). However, similar results were found for tests between empirical groups (electronic supplementary material, tables S2–S3). Although our simulations are designed to constrain different LBG types, the steepness of these gradients is also reliant on the distribution of available shallow marine area for populating with occurrences, producing more realistic distributions for each time interval (i.e. a species–area relationship [41]). For instance, if a latitudinal band has twice as many cells as another latitudinal band, but the cells in those bands have the same weighted probability, the former is likely to be much more diverse than the latter.

## (b) Spatial sampling coverage and extent

To test how spatial sampling influences our understanding of the LBG through time, we sampled our simulated occurrence data using the palaeogeographic sampling window of real fossil collections for each stratigraphic stage. To generate our per-stage sampling window, we downloaded all available Asselian–Piacenzian marine collection data from the PBDB (www.paleobiodb.org) on the 28 September 2020, yielding 108 108 fossil collections. Using the palaeo-coordinates of collections (rotated using the GPlates plate model [42]), we produced global stage-level grids ($1° \times 1°$) of sampled area at the same extent of our DEMs, such that a grid cell would be deemed to have been sampled if at least one collection appeared within it. Our sampled grids were then filtered to include only cells falling within our shallow marine area grids.

For the purposes of this study, we establish a metric that we term 'spatial sampling coverage' (SSC) to assess whether the available sampled area controls biodiversity estimates between stages and palaeolatitudinal bins. Here, SSC is defined as the percentage of sampled cells (cells containing collections) within a chosen set of available cells (total number of cells within the shallow marine grid). As we were also interested in the spatial extent of sampling in the fossil record, we calculated the summed minimum-spanning tree (MST) length, the minimum total distance of segments capable of connecting all sampled cells (cells containing at least one collection), using the spantree() function from the R package 'vegan' [43]. We calculated the SSC and summed MST length for each stage at both a global scale, and within 15° palaeolatitudinal bins, for comparison with biodiversity estimates.

## (c) Latitudinal biodiversity gradient analyses

To test the extent to which relative biodiversity patterns can be recovered from degraded samples, we computed the mean richness for simulated, sampled, and sampling-standardized LBGs for each stage. All LBGs were computed within 15° palaeolatitudinal bins, a common bin size used in deep time studies (e.g. [14,26]). For sampling-standardization, we applied classical rarefaction to the sampled occurrence data to account for large differences in sampling between palaeolatitudinal bins [44]. We kept the number of occurrences drawn from each palaeolatitudinal bin at 50 occurrences, a similar threshold to previous fossil biodiversity studies [13,45]. Sampling-standardization was implemented with 1000 bootstrap replicates for each LBG simulation replication/type ($100 \times 3$), with the mean richness recorded for each palaeolatitudinal bin. As we were principally interested in the relative shape of LBGs, we normalized all (simulated, sampled and sampling-standardized) LBGs within their respective stages on a scale from 0 to 1 to produce proportional richness curves. This was achieved by dividing the richness of each palaeolatitudinal bin by the sum of richness across palaeolatitudinal bins for each respective stage. Recovering 'true' levels of biodiversity in the fossil record is an impossible challenge; however, the use of our proportional richness curves provides an accurate understanding of the relative latitudinal distribution of biodiversity.

To quantify the similarity between simulated biodiversity curves and sampled/sampling-standardized counterparts, we: (i) calculated 'model residuals' by subtracting simulated biodiversity curves from their sampled/sampling-standardized counterparts to determine in which palaeolatitudinal bins associated richness is over- or under-represented; (ii) computed correlations (Pearson's correlation coefficient $r$) between pairwise combinations of simulated, sampled and sampling-standardized biodiversity curves for each stage and LBG type; and (iii) quantified the total displacement ($D$) between pairwise combinations of simulated, sampled and sampling-standardized biodiversity curves for each stage and LBG type using:

$$D = \sum_{j=1}^{n} d_j,$$

where $d$ represents the absolute (i.e. the non-negative value) difference in richness between curves for each palaeolatitudinal bin for $j = 1, 2, \ldots, n$ palaeolatitudinal bins. This metric provides a measure of similarity between proportional (each palaeolatitudinal bin divided by the sum of palaeolatitudinal bins) richness curves where $D$ values can range between 0 and 2, with 0 representing no difference between pair-wise combinations of biodiversity curves, and 2 indicating maximum possible difference. However, as different types of LBG might have inherent overlap owing to the distribution of simulated biodiversity, our comparisons of simulated LBGs provide a null expectation of the difference between LBG types, against which sampled and sampling-standardized LBGs can be evaluated. In addition, we used Kolmogorov–Smirnov two-sample tests to evaluate the detectability of different types of LBG after filtering by the spatial sampling window of fossil collections. To do so, we compared simulated biodiversity curves with their sampled/sampling-standardized counterparts. Prominent peaks in palaeolatitudinal richness (e.g. a temperate peak) are typically used to infer the shape of LBGs in deep time (e.g. [13,29]). Therefore, we also determined the palaeolatitudinal bin with peak biodiversity for simulated, sampled and sampling-standardized LBGs to identify whether

*Proc. R. Soc. B* **288**: 20202762

limited spatial sampling shifted the observed peak in biodiversity between palaeolatitudinal bins and zones: tropics (0–30° N/S), temperate (30–60° N/S) and polar (60–90° N/S).

To provide insight into the impact of heterogeneous spatial sampling, we calculated the coefficient of determination ($R^2$) between species richness (simulated, sampled and sampling-standardized) and SSC/summed MST length for each palaeolatitudinal bin and LBG type. Specifically, these tests determine whether palaeolatitudinal sampling drives observed biodiversity trends, and hence the detectability of different types of LBG.

Finally, for each of our LBG types, we also computed the sampled global richness to assess whether variation in our spatial sampling window impacted upon sampled global biodiversity for different types of LBGs. We did so by calculating the coefficient of determination ($R^2$) between global sampled richness and global SSC/summed MST length for each LBG type.

## 3. Results

### (a) Spatial sampling coverage and extent

Analyses of fossil collections indicate palaeogeographic SSC is heterogeneous through both time and space. Global SSC does not exceed 2% for any stage, with a mean global SSC of 0.70% (±0.32% standard deviation) across all 56 geological stages spanning approximately 300 Myr (electronic supplementary material, figure S7a). Palaeolatitudinal SSC is relatively high for palaeolatitudinal bin 30–45° N with a mean SSC of 2.68% across all geological stages (figure 2a). However, all other palaeolatitudinal bins have a mean SSC lower than 1.5%, indicating unevenness in palaeolatitudinal sampling (figure 2a). In addition, there is notable temporal variability within palaeolatitudinal bins. For example, palaeolatitudinal bin 30–45° N has an SSC of 1.20% for the Barremian (Cretaceous), whereas the subsequent stage (Aptian, Cretaceous) has an SSC of 3.76% (figure 2a). Furthermore, approximately 40% of palaeolatitudinal bins have an SSC of 0%, indicating substantial data absence along our time series. Overall, SSC is generally skewed towards 30–45° N from the Late Triassic (Norian) to the Neogene (Piacenzian) (figure 2a).

The palaeogeographic spread of sampled shallow marine area, measured by summed MST length, indicates similar temporal and spatial heterogeneity to SSC. Global summed MST length has a mean value of 44 936 km, with a standard deviation of 12 308 km, across all 56 stratigraphic stages (electronic supplementary material, figure S7b). Mean palaeolatitudinal summed MST length is highest for palaeolatitudinal bin 30–45° N at 14 865 km, and lowest for 75–90° S at just 365 km. All other palaeolatitudinal bins range between 1009 and 10 921 km (figure 2b). Temporal variability in the palaeogeographic spread of sampling within palaeolatitudinal bins is also substantial. For example, the palaeolatitudinal bin 30–45° N has a summed MST length of 9168 km for the Ladinian, which is more than double that of the former stratigraphic stage (Anisian: 3650 km) (figure 2b). Furthermore, approximately 50% of palaeolatitudinal bins have a palaeogeographic sampling spread of 0 km (owing to either 1 or no cells being sampled in the bin).

### (b) Latitudinal biodiversity gradient analyses

Broad changes between simulated palaeolatitudinal biodiversity curves and their sampled/sampling-standardized counterparts can be observed in figure 3 (stage-level plots

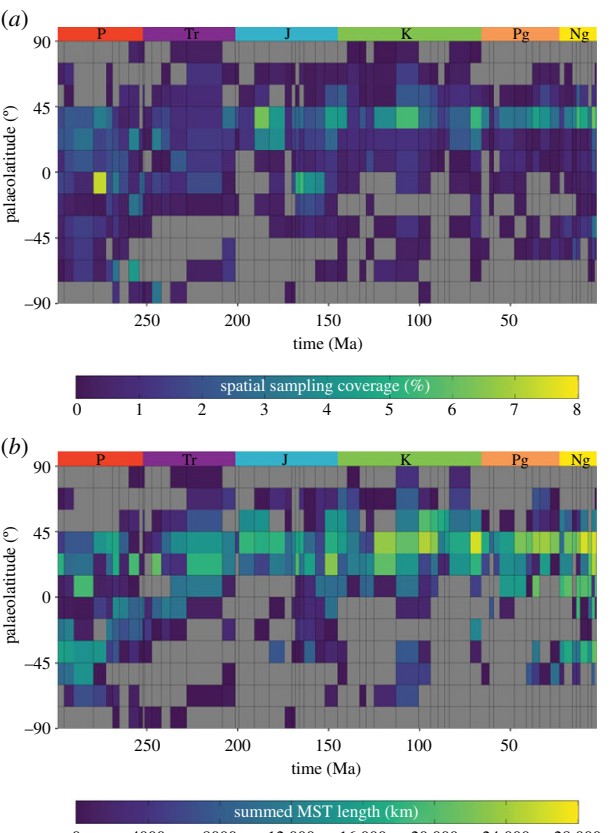

**Figure 2.** Spatial sampling coverage (SSC) (*a*) and extent (*b*) for shallow marine environments in the Asselian–Piacenzian (298.9–2.58 Ma; Permian–Neogene) fossil record. (*a*) SSC is calculated as the percentage of sampled cells within available cells, where available cells are those within shallow marine grids (less than or equal to 200 m). (*b*) Summed minimum spanning tree (MST) length (km) is calculated as the minimum total length of all segments connecting sampled cells within shallow marine environments. Grey tiles (*a*–*b*) illustrate palaeolatitudinal bins with values of 0 for ease of identification. Sampling metrics (SSC and MST) are computed within 15° palaeolatitudinal bins for each stratigraphic stage and were calculated at a spatial resolution of 1° × 1° (approx. 110 × 110 km² at the Equator). Period abbreviations are as follows: Permian (P), Triassic (Tr), Jurassic (Jr), Cretaceous (K), Palaeogene (Pg) and Neogene (Ng). (Online version in colour.)

are included in the electronic supplementary material 2). Analyses of model residuals indicate that biodiversity is frequently over-represented at temperate palaeolatitudes in the Northern Hemisphere (30–45° N), after filtering occurrences by the palaeogeographic sampling window of fossil collections (electronic supplementary material, figure S8). Conversely, palaeolatitudinal bins in the tropics frequently have their associated species richness under-represented, as does most of the Southern Hemisphere (electronic supplementary material, figure S8). Sampling-standardization tends to reduce model residuals for palaeolatitudinal bins with sufficient spatial sampling coverage, but has no effect on the approximately 40% of bins suffering from a total lack of sampling.

The total displacement between pairwise combinations of latitudinal biodiversity gradients provides insight into the similarity between simulated, sampled and sampling-standardized biodiversity curves (electronic supplementary material, figures S9–S10). Overall, displacement scores suggest sampled and sampling-standardized biodiversity curves tend to be a

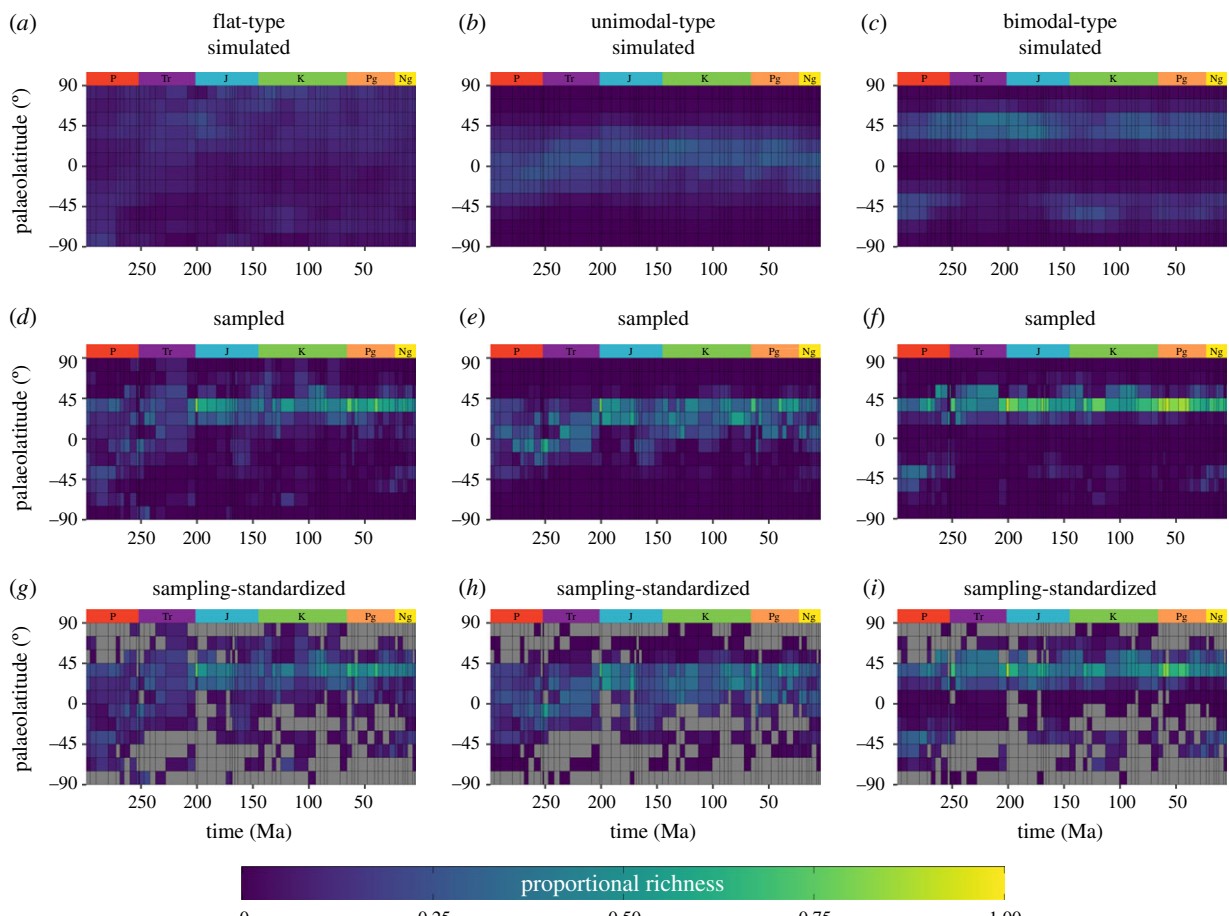

**Figure 3.** Heat maps of mean simulated (*a*–*c*), sampled (*d*–*f*) and sampling-standardized (classical rarefaction) (*g*–*i*) richness from the Asselian (298.9 Ma, Permian) to Piacenzian (2.58 Ma, Neogene) at stage-level, and within 15° palaeolatitudinal bins. Palaeolatitudinal richness values within each stage (*a*–*i*) represent their proportional richness (palaeolatitudinal bin richness divided by the sum of palaeolatitudinal bins). Grey tiles (*g*–*i*) illustrate palaeolatitudinal bins with insufficient data to compute rarefied richness estimates. For each type of latitudinal biodiversity gradient, simulated richness values (*a*–*c*) are based on the average bin value from the 100 replications. Sampled richness values (*d*–*f*) are based on the average bin value from the 100 sampled replications. Rarefied palaeolatitudinal richness estimates (*g*–*i*) are based on the mean of means of the 1000 bootstrap replicates from rarefaction for each replication of latitudinal biodiversity gradient simulations (=100). Period abbreviations are as follows: Permian (P), Triassic (Tr), Jurassic (Jr), Cretaceous (K), Palaeogene (Pg) and Neogene (Ng). (Online version in colour.)

poor representation of their 'true' (simulated) gradient (electronic supplementary material, figure S9). Furthermore, simulated LBG types may converge after sampling (electronic supplementary material, figure S10). Simulated unimodal- and bimodal-type LBGs maintain a fairly constant displacement ($\mu = 1.27$, $\sigma = 0.07$) along the entire time series (figure 4). However, after filtering by the spatial sampling window of fossil collections, displacement scores between sampled unimodal- and bimodal-type LBGs ($\mu = 1.04$, $\sigma = 0.26$) are significantly lower than those of simulated LBGs, indicating increased similarity between biodiversity curves ($p < 0.001$; Wilcoxon rank sum test). Notably, the displacement between unimodal- and bimodal-type biodiversity curves is substantially lowered for numerous stages after sampling; for example, the displacement between Hettangian LBGs is reduced from 1.22 to 0.35 (figure 4), and the resulting sampled biodiversity curves appear virtually identical, regardless of the underlying simulated distribution of biodiversity (electronic supplementary material 2: Hettangian). This is particularly evident in the following stages: Hettangian–Pliensbachian (Jurassic), Santonian (Cretaceous), Danian, Bartonian, and Chattian (Palaeogene) (figure 4; electronic supplementary material, S2). Conversely, displacement scores between sampled unimodal- and bimodal-type LBGs are comparatively

high for most of the Permian, Triassic and Neogene (figure 4), making it possible to distinguish between simulated LBG types (see stage-level plots in the electronic supplementary material, S2). These findings are supported by the relatively low displacement scores found between unimodal-type simulated and sampled LBGs, as well as bimodal-type simulated and sampled LBGs (electronic supplementary material, figure S9). Sampling-standardization tends not to improve upon these observed trends, with displacement scores between sampling-standardized unimodal- and bimodal-type LBGs ($\mu = 0.93$, $\sigma = 0.22$; figure 4) significantly lower than those from sampled LBGs ($p < 0.01$; Wilcoxon rank sum test).

We calculated Pearson's correlation coefficient (*r*) between pairwise combinations of LBGs to further test the similarity between palaeolatitudinal biodiversity curves. Mean Pearson's *r* scores suggest that sampled (flat: 0.372; unimodal: 0.722; bimodal: 0.743) and sampling-standardized (flat: 0.460; unimodal: 0.721; bimodal: 0.865) LBGs are relatively representative of their 'true' (simulated) underlying gradient (electronic supplementary material, figure S11). However, the standard deviation of Pearson's *r* scores demonstrates that this is highly variable across stratigraphic stages for both sampled (flat: 0.211; unimodal: 0.166; bimodal: 0.153) and sampling-standardized (flat: 0.286; unimodal: 0.253; bimodal: 0.117)

**Figure 4.** Total displacement (*D*; see Methods) between unimodal- and bimodal-type latitudinal biodiversity gradients (Asselian to Piacenzian). The dark blue line indicates the total displacement between proportional simulated unimodal- and bimodal-type latitudinal biodiversity gradients. This provides an expected difference between the two different types of latitudinal biodiversity gradient. The blue line indicates the total displacement between sampled unimodal- and bimodal-type latitudinal biodiversity gradients. The light blue line depicts the total displacement between sampling-standardized unimodal- and bimodal-type latitudinal biodiversity gradients. The smaller the total displacement, the more similar the diversity curves. Notably low displacement scores are observed for the Hettangian–Pliensbachian (Jurassic), Santonian (Cretaceous), Danian, Bartonian and Chattian (Palaeogene) for sampled and sampling-standardized latitudinal biodiversity gradients. These stages are highlighted because unimodal- and bimodal-type latitudinal bio-diversity gradients are nearly identical after sampling (see the electronic supplementary material, S2). Period abbreviations are as follows: Permian (P), Triassic (Tr), Jurassic (Jr), Cretaceous (K), Palaeogene (Pg) and Neogene (Ng). (Online version in colour.)

LBGs (electronic supplementary material, figure S11). Furthermore, Pearson's *r* scores are generally insignificant ($p \geq 0.05$) for both sampled and sampling-standardized LBGs (electronic supplementary material, figure S11). The ability to recover different simulated LBG types is variable across the time series. For example, simulated and sampled/sampling-standardized biodiversity curves are generally highly correlated for unimodal- and bimodal-type LBGs during the Permian and Triassic. However, for flat-type LBGs, simulated and sampled/sampling-standardized biodiversity curves tend to be weakly correlated in the Permian (electronic supplementary material, figure S11). Pairwise comparisons between types of LBG suggest sampled flat-, unimodal- and bimodal-type diversity curves are all significantly ($p < 0.001$; Wilcoxon rank sum test) more correlated to one another, than their 'true' simulated counterpart (electronic supplementary material, figure S12). This trend is also observed for pairwise comparisons between simulated and sampling-standardized biodiversity curves ($p < 0.001$; Wilcoxon rank sum test). Furthermore, sampled biodiversity curves for all LBG types are notably more correlated to one another in the Jurassic–Palaeogene than the Permian and Triassic (electronic supplementary material, figure S12). Finally, Pearson's *r* scores between different types of sampled/sampling-standardized LBGs tend to be statistically significant ($p < 0.05$), while simulated are not (electronic supplementary material, figure S12).

We evaluated the detectability of LBGs in deep time by comparing simulated biodiversity curves with their sampled/sampling-standardized counterparts using Kolmogorov–Smirnov two-sample tests. Based on this analysis, we found that 25 out of 56 (approx. 45%) of flat-, 12 out of 56

(approx. 21%) of unimodal-, and 23 out of 56 (approx. 41%) bimodal-type sampled LBGs were statistically different ($p < 0.05$) from their simulated counterparts (electronic supplementary material, figure S13). At face value, sampling-standardization would appear to generally improve upon this with only 17 out of 56 (approx. 30%) of flat-, 10 out of 56 (approx. 18%) of unimodal- and 21 out of 56 (approx. 38%) bimodal-type LBGs statistically different from their simulated counterparts ($p < 0.05$). However, these results are influenced by the presence of palaeolatitudinal bins without richness estimates (i.e. reducing sample size).

Computation of the palaeolatitudinal zone with peak richness indicates that sampled flat- and unimodal-type LBGs would have their genuine zonal (tropics, temperate or polar) peak in biodiversity recovered in approximately 30% and 73% of stages, respectively (electronic supplementary material, table S4). When considering the exact palaeolatitudinal bin with peak biodiversity, these figures are reduced to approximately 18% for flat- and approximately 48% for unimodal-type LBGs (electronic supplementary material, table S4; figure S14). However, for bimodal-type LBGs, the exact palaeolatitudinal bin with peak richness would be recovered for approximately 70% of stages, and approximately 96% when considering the correct palaeolatitudinal zone (i.e. temperate regions). Distinguishing between types of LBG after sampling is often not possible with both tropical (unimodal-type LBG) and temperate (bimodal-type LBG) peaks in biodiversity only identifiable in approximately 70% stages. Sampling-standardization only minorly improves upon these observations (electronic supplementary material, table S4). Notably, palaeolatitudinal peaks in richness are regularly shifted to 30–45° N after filtering by the spatial sampling window of fossil collections, regardless of the underlying distribution of biodiversity (electronic supplementary material, figure S14).

We tested the relationship between palaeolatitudinal biodiversity (simulated, sampled and sampling-standardized LBGs) and spatial sampling metrics (SSC and summed MST length). Overall, coefficient of determination test results suggest stronger relationships between sampled/sampling-standardized palaeolatitudinal richness and sampling metrics, than simulated palaeolatitudinal richness and sampling metrics (electronic supplementary material, figure S15–S16). A moderate to strong correlation was found between sampled flat- ($R^2 = 0.788$, $p < 0.001$), unimodal- ($R^2 = 0.492$, $p < 0.001$), and bimodal-type ($R^2 = 0.431$, $p < 0.001$) palaeolatitudinal richness and SSC (electronic supplementary material, figure S15). A weaker, but still significant, relationship was also observed between sampled LBGs and summed MST length (electronic supplementary material, figure S16): flat- ($R^2 = 0.471$, $p < 0.001$), unimodal- ($R^2 = 0.134$, $p < 0.001$), and bimodal-type ($R^2 = 0.295$, $p < 0.001$). Sampling-standardized LBGs indicate similar relationships between palaeolatitudinal richness and spatial sampling metrics, and are therefore not discussed further (electronic supplementary material, figures S15–S16).

## (c) Global biodiversity

Global richness counts from the sampled flat- ($\mu = 79.90$; $\sigma = 23.93$), unimodal- ($\mu = 119.18$; $\sigma = 27.53$) and bimodal-type ($\mu = 128.08$; $\sigma = 57.81$) LBGs express temporal differences in biodiversity through time, suggesting observed extinction and radiation events might be influenced by both underlying

patterns of biodiversity and sampling (electronic supplementary material, figure S17). Coefficient of determination results suggest that global SSC explains a large proportion of the variance observed in global richness for sampled flat- ($R^2 = 0.941$, $p < 0.001$), unimodal- ($R^2 = 0.659$, $p < 0.001$), and bimodal-type ($R^2 = 0.417$, $p < 0.001$) LBGs (electronic supplementary material, figure S18). Similar results are observed between global sampled diversity and summed MST length (electronic supplementary material, figure S18): flat- ($R^2 = 0.503$, $p < 0.001$), unimodal- ($R^2 = 0.262$, $p < 0.001$), and bimodal-type ($R^2 = 0.442$, $p < 0.001$). However, SSC appears to generally explain more variance in sampled global diversity than the spread of sampling (summed MST length).

## 4. Discussion

Our results show that distinguishing between different types of LBG in deep time can be problematic owing to the limited spatial sampling window. Spatio-temporal analyses indicate that LBGs reconstructed from the fossil record are a poor representation of the true distribution of past biodiversity, at least in the shallow marine realm, with simulated LBGs and their sampled counterparts notably different (figure 3). This observation is especially clear from our stage-level results: (i) model residuals suggest proportional richness is frequently over-represented at temperate latitudes in the Northern Hemisphere (30–60° N), and under-represented in the tropical latitudes, as well as the Southern Hemisphere; (ii) displacement scores ($D$) indicate considerable difference between simulated LBGs and their sampled counterparts for all LBG types; (iii) Pearson's correlation coefficient ($r$) scores demonstrate spatio-temporal variability in the relationships between simulated and sampled LBGs; (iv) Kolmogorov–Smirnov two-sample tests suggest statistically significant differences between simulated LBGs and their sampled counterparts; and (v) analyses of the palaeolatitudinal zone (tropics, temperate or polar regions) with peak richness suggests that sampled flat- and unimodal-type LBGs frequently have peaks in diversity within different palaeolatitudinal zones to their simulated counterparts.

The same pattern of signal degradation following sampling is observed when comparing LBG types. While our results generally indicate non-significant low to moderate correlations between different types of simulated LBG (flat-, unimodal- and bimodal-type), sampled pairwise comparisons are frequently significant, and highly correlated (electronic supplementary material, figure S12). This finding is supported by displacement scores between pairwise combinations of different types of simulated LBGs, which indicate relatively sustained differences throughout the time period of study (figure 4 and electronic supplementary material, figure S10). However, after applying the spatial sampling window of fossil collections, the displacement between sampled LBGs is significantly reduced from their simulated counterparts. Consequently, despite different empirical distributions of biodiversity, relative biodiversity patterns might converge after sampling. As a result, differentiating between types of sampled LBGs can be problematic; for example, in approximately 30% of stages we cannot accurately identify both tropical and temperate peaks in biodiversity (electronic supplementary material, figure S14). Our results corroborate previous studies (e.g. [11,24,46]) in suggesting that deep time macroecological

patterns have the capacity to be strongly shaped by the available spatial sampling window, as opposed to the genuine distribution of biodiversity (figure 3). Considering that our sampling regime (all occurrences within a sampled cell are preserved and selected) is an impossible 'best case' scenario for the fossil record, it is likely that the impacts of variable spatial sampling are more severe than reported here.

We support previous findings that demonstrate a temporal shift in palaeolatitudinal sampling from the Permian towards the present-day [4,37,47]. In general, this shift follows a poleward drift from relatively equable palaeolatitudinal sampling in the Permian and Triassic, to temperate palaeolatitudes in the Jurassic to Palaeogene (figure 2). While the spatial extent of sampling is more equable in the Neogene, spatial sampling coverage is still skewed towards 30–45° N (figure 2). Remarkably, the skewness in palaeolatitudinal spatial sampling towards 30–45° N is well represented by our LBG simulations (figures 2 and 3). While approximately 70% of the genuine palaeolatitudinal peaks in richness would be detected for bimodal-type LBGs, only approximately 18% of flat- and 50% of unimodal-type LBGs would have their genuine peak in diversity observed in their correct palaeolatitudinal bin (electronic supplementary material, table S4). Regardless of the simulated distribution of biodiversity, palaeolatitudinal peaks frequently converge on 30–45° N after applying the spatial sampling window of fossil collections (electronic supplementary material, figure S14). Although this is largely related to the palaeo-location of the USA, China and Europe (i.e. regions of high sampling intensity) [48,49], the presence of major epicontinental seas (e.g. Laurasian and Western Interior seaways) in the Northern Hemisphere during the Mesozoic enhanced the available area for shallow marine faunas, contributing to the heterogeneous spatial distribution of shallow marine fossils [50–52]. This is supported by our simulated flat-type LBGs, which simulate high Mesozoic biodiversity in the Northern Hemisphere (electronic supplementary material, figure S14).

Sampling-standardization has become a commonplace methodology in palaeontological studies for reducing the impact of fossil record bias upon diversity curves [9,14,29,30,32,34]. Our results indicate that while sampling-standardization alleviates some issues of data quality, it is often confounded by spatial sampling variability. When broken down by LBG type, our simulation results show that sampling-standardization tends to decrease the displacement between simulated and sampled LBGs, improving relative palaeolatitudinal richness ratios (electronic supplementary material, figure S9). This improvement is more prominent in bimodal-type LBGs than unimodal-type LBGs, as a direct result of the close agreement between simulated bimodal-type LBGs and the spatial sampling window along our time series (figures 2 and 3; electronic supplementary material, figure S9). However, despite this improvement, sampling-standardization generally decreases the total displacement between unimodal- and bimodal-type LBGs across the entire time series, therefore increasing similarity between LBG types (figure 4). This is further supported by sampling-standardization failing to differentiate between artificial similar palaeolatitudinal peaks in richness between different types of LBG (electronic supplementary material, figure S14). In addition, sampling-standardized LBGs tend to maintain moderate to strong positive correlations (Pearson's $r$) between

pairwise combinations of LBG type (electronic supplementary material, figure S12). However, this shortfall is not the result of the elected sampling-standardization approach, but the available data. Numerous palaeolatitudinal bins (approx. 40%) in our study period have zero SSC, and hence zero species occurrences after applying the available spatial sampling window, which prevents the computation of rarefied biodiversity estimates. Owing to this data absence, as well as preferential sampling between 30–45° N during the Mesozoic, sampling-standardized LBGs frequently identify a 30–45° N peak in biodiversity (stage-level plots available within the electronic supplementary material, S2). It is possible that other sampling-standardization approaches such as shareholder quorum subsampling [33] and extrapolators (e.g. Chao 2 [53]) might perform better in estimating relative biodiversity patterns, when sufficient data are available [32]. However, as approximately 40% of palaeolatitudinal bins suffer from complete data absence, which no sampling-standardization approach can account for, it is unlikely to improve upon the detectability of genuine LBGs in deep time.

Our results additionally allow us to identify particular time intervals that might be well or poorly suited for reconstructing LBGs in deep time. The reconstruction of different types of LBG appears to be particularly challenging for the Hettangian–Pliensbachian (Jurassic), Santonian (Cretaceous), Danian, Bartonian and Chattian (Palaeogene) (figure 4). During these intervals, simulated unimodal- and bimodal-type LBGs are almost identical after applying the spatial sampling window (see the electronic supplementary material, S2 for individual stage-level plots). This limitation is driven by skewed palaeolatitudinal sampling towards the Northern Hemisphere (figure 2). However, it should also be noted that when skewness in palaeolatitudinal sampling is more equable, the reconstruction of different types of LBG is possible, such as during the Permian and Early Triassic (figure 2–4). Nevertheless, skewed palaeolatitudinal sampling may be negated at higher SSC owing to latitudinal differences in species discovery curves (e.g. if diversity is genuinely greater at lower latitudes, more species might be discovered in smaller sampled areas here, than larger sampled areas at higher latitudes).

Although these issues are mostly a concern for spatial studies, the temporal component of biodiversity is also strongly affected. Our global biodiversity curves, constructed from sampled LBGs, support the hypothesis that palaeolatitudinal sampling bias impacts upon our understanding of phenomena such as extinction and radiation events [4] (electronic supplementary material, figure S17). As shown above, a large proportion of our understanding of the Mesozoic and Cenozoic fossil record is sourced from a single palaeolatitudinal band (30–45°) within the Northern Hemisphere [4,37]. If the LBG has varied through time, with shifts between temperate and tropical latitudinal peaks, this could add volatility to global biodiversity curves that have not accounted for palaeolatitudinal SSC [11]. If sampling remains focused on one palaeolatitudinal bin through time, a genuine alternation in the LBG from a temperate to tropical peak could instead appear in the fossil record as a major extinction or radiation event [4,25].

It should be noted that our study is not without limitations. Firstly, our models are simple, with biodiversity patterns only driven by our specified type of LBG, with the steepness of these gradients influenced by the available shallow marine area (cells), emulating the species–area relationship. As such,

they are not influenced by other factors that are probably crucial to the formation of the LBG, such as climate [54,55]. Although both abiotic and biotic drivers have been hypothesized to contribute to the formation of LBGs, here we are principally interested in whether different relative latitudinal biodiversity patterns can be reconstructed after applying the 'known' sampling window, rather than reconstructing 'actual' biodiversity patterns in deep time. The results presented here might also be dependent upon our sampling protocol in which we sampled all occurrences (and hence biodiversity) within a cell providing it contains at least one collection. This simplified approach ignores the relationship between the number of samples (i.e. collections) and observed richness (i.e. richness-sample accumulation curve). However, using a weighted approach based on the number of collections within cells to sample occurrences would only further support the trends reported here, owing to the concentration of collections between 30–45° N for the majority of our time series. The results presented here might also be dependent upon the DEMs and palaeorotations used in this study, necessitated by data availability. While analyses using different versions of DEMs and palaeorotation models might produce different results to those presented here, it is likely that they would not change the general trends observed in this study. As our simulated occurrence data emulate those of modern marine invertebrates, there may be differences in the range size and occurrence frequency distributions of fossil and simulated taxa owing to time averaging effects. However, as fossil occurrence data is inherently biased by a number of anthropogenic and geological filters, approximating range size and frequency distributions from these data is likely to be a poorer approximation of these attributes than modern occurrence data. Finally, our study is limited by design to the shallow marine realm. While the impact of variable spatial sampling on estimates of terrestrial diversity is well documented [11,24,56,57], it is likely that reconstructing LBGs on land is also challenging as the terrestrial fossil record is generally less complete and abundant than the marine fossil record [49]. Constraining the impact of spatial sampling heterogeneity on terrestrial, as well as marine systems, is imperative as the LBG might respond differently on land compared to the marine realm [29,30,58]. In particular, terrestrial LBGs might be influenced by different drivers (e.g. precipitation [54]), which could in turn influence their relationship with sampling.

Our study highlights the impact of spatial bias on our understanding of past biogeographic patterns, identifying the potential issues of applying sampling-standardization methods without considering the heterogeneity of global sampling. While our results show the application of such methods is not meaningless, they do highlight key issues with current approaches for reconstructing macroecological patterns in deep time. Specifically, these methods cannot account for artefactual data absence, which we show can lead to erroneous conclusions. The advent of large, research community-driven palaeontological databases has substantially improved the availability of presence/occurrence information. However, few publicly accessible records succeed in documenting and differentiating between sites that have been sampled with no fossil occurrences found (sampled absence), versus sites that have not been sampled and therefore do not record fossil occurrences (potentially false absences). Inferential methods commonly used in ecology, such as ecological niche and occupancy modelling might therefore represent an

alternative approach to identify where we might be missing data, although they remain in their infancy in palaeobiological studies [59–62]. While these methods cannot replace prospecting for fossil sites, or inform us directly about biodiversity, they allow us to infer what data we might be missing, and from where. However, we contend that only additional sampling within relatively poorly sampled regions will provide empirical evidence on past latitudinal biodiversity patterns. Collecting fossil data from such under-sampled regions provides an opportunity to improve our understanding of the evolution and distribution of life on Earth, and will ultimately help to resolve many of the current inadequacies highlighted in this study. Overall, our results suggest caution should be exercised when attempting to reconstruct LBGs in deep time, with current occurrence-based methods susceptible to the impact of heterogeneity in spatial sampling.

## 5. Conclusion

Our simulated LBGs suggest that the current distribution of sampled area of shallow marine fossils frequently hinders the differentiation between types of LBG. A long-term poleward shift in the palaeolatitudinal sampling window is observed, which enables different types of LBGs to be readily identified in the Permian and Triassic when palaeolatitudinal sampling was less skewed towards temperate palaeolatitudes. The reconstruction of unimodal gradients (tropical peaks) in the Jurassic, Cretaceous and Palaeogene is limited owing to skewed sampling towards northern temperate regions. However, more equable palaeolatitudinal sampling in the Neogene permits the recovery of unimodal gradients during this time period. Sampling-standardization aids reconstruction of relative biodiversity patterns when palaeolatitudinal spatial sampling coverage is not severely skewed. However, reconstruction of deep time LBGs is severely impacted by artefactual absences, with 40% of palaeolatitudinal bins suffering from complete data absence that prevents computation of sampling-standardized biodiversity estimates. Consequently, previous studies might therefore have failed to recover the 'true' LBG type owing to the heterogeneous distribution of the available sedimentary record and incomplete sampling.

Data accessibility. All electronic supplementary material and data have been included as part of the submission. All simulations and analyses were performed in R v. 3.6.2, and are available on GitHub (accessible via: https://github.com/LewisAJones/LBG_sim).

Authors' contributions. All authors conceived and designed the project; L.A.J. developed and performed the analyses; L.A.J., C.D.D. and P.D.M. interpreted the data; all authors contributed to the writing of the manuscript; L.A.J. and C.D.D. produced the figures.

Competing interests. We declare we have no competing interests.

Funding. L.A.J. was supported by an Imperial College London President's PhD Scholarship. C.D.D.'s contribution was funded by the European Union's Horizon 2020 research and innovation programme 2014–2018 under grant agreement no. 637483 (ERC Starting Grant TERRA to Richard J. Butler). P.D.M.'s contribution was supported by a Royal Society University Research Fellowship (grant no. UF160216). A.F.'s contribution was supported by grants from the Natural Environmental Council Research (grant nos NE/I005714/1 and NE/P013805/1).

Acknowledgements. We would like to thank Danielle Fraser, Matthew Powell and six other anonymous reviewers for their helpful comments on previous versions of this manuscript that greatly improved this work. We are also grateful for the efforts of all those who have collected and entered data into the Ocean Biogeographic Information System and the Paleobiology Database. We would further like to thank Getech Group PLC for providing the digital elevation models used in this study. This is Paleobiology Database official publication number 391.

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
