## [Peer Review File · Proceedings of the Royal Society B: Biological Sciences]

Review History

RSPB-2020-2762.R0 (Original submission)

Review form: Reviewer 1

Recommendation

Accept with minor revision (please list in comments)

Scientific importance: Is the manuscript an original and important contribution to its field?

Good

General interest: Is the paper of sufficient general interest?

Good

Quality of the paper: Is the overall quality of the paper suitable?

Excellent

Is the length of the paper justified?

Yes

Should the paper be seen by a specialist statistical reviewer?

No

Do you have any concerns about statistical analyses in this paper? If so, please specify them explicitly in your report.

No

It is a condition of publication that authors make their supporting data, code and materials available - either as supplementary material or hosted in an external repository. Please rate, if applicable, the supporting data on the following criteria.

Is it accessible?

Yes

Is it clear?

Yes

Is it adequate?

Yes

Do you have any ethical concerns with this paper?

No

Comments to the Author

The manuscript by Jones et al., on the extent to which sampling bias influences our perception of spatial biodiversity patterns in deep time, is an excellent contribution. The analyses are well considered and thorough, and the manuscript is clearly written and accessible given the complexity of the methodology. I consider it to fit well within the remit of Proceedings B, and suggest only minor changes to the manuscript, as outlined below, prior to publication.

L20 - Maybe replace 'indistinguishable' with 'obscured'.

L26 - I would replace 'obscure' here with 'prevent' or 'inhibit'.

L32 - Might be worth adding the qualifier 'future climate change', or similar ('anthropogenic?').

L37 - The first question here is somewhat unnecessary.

L39 - I'm unsure about the word 'grain', perhaps 'scale' and/or 'resolution' (unless you meant something else?) would be more accurate.

L41 - This is pedantic (!) but 'decreases' implies a directional or linear variable, which 'regions' aren't strictly, so despite a possible reluctance to repeat 'latitudes' I think that would be better, or another similar rewording.

L43 - Replace 'it', perhaps 'large-scale spatial variation in biodiversity'.

e.g. L53 - I think you should cite Antell et al. (doi.org/10.1016/j.cub.2019.10.065) as another example considering spatial sampling bias in the marine fossil record.

L58 - 'taxon pool' feels ambiguous between a pool of occurrences and the taxa counted within it. You could be explicit about both, i.e. variation in geographic area -> variation in sample size of occurrences -> strong control on variation in taxon counts.

L62 - You could remove 'in data originating'.

L63 - Change to e.g. '(flat, unimodal and bimodal) using palaeogeographies of 56'.

L64 - Why did you use this time interval?

L68-9 - I find this sentence confusing. Both simulated and sampled curves were produced based on the DEMs, but surely only the sampled curve was subjected to the sampling window filter?

The second half of the sentence then seems to contradict your use of 'known' in L74.

L78 - It would be more accurate to specify 'life' as 'marine invertebrate' here.

L95 - Could you give a range (or average) of range sizes?

L98 - I think 'We did so...' is an unnecessary sentence.

L99-102 - I think this interpretation could be explained in a way which is more understandable, for example 'Our simulated species had range sizes and frequency distributions which replicated those of sponges and echinoderms, but were less similar to those of bivalves and anthozoans' (if I've understood your table correctly?). And/or you could change the sentiment in L101-2 to something along the lines of 'Empirical data from these clades show that they vary in their range sizes and occurrence frequency distributions, and our intention was to simulate a mixed

community of these animals rather than to precisely replicate any one of them, therefore the simulation performs well". You could probably write this more succinctly than I have! Otherwise, change 'both' to 'a mixture of', or similar, and add 'simulated and empirical groups, and the metric used to compare them'.

L101 - Perhaps a point to add to the discussion, if not here: do you think differences in the amount of time averaging between your fossil data and OBIS observations could make them an unfair comparison for range sizes and frequency distributions?

L103-111 - I think this paragraph should be inserted into the one above, around L88, to save coming back to how the probability grids were used.

L110 - Change 'between' to 'in those'.

L114 - Add 'real fossil collections'.

L116 - Was this "all marine", or literally everything? If everything, could the inclusion of terrestrial collections in coastal grid cells have influenced your sampling masks?

L117-120 - Were these the palaeo-coordinates produced by the PBDB's model, i.e. the GPlates model? If your DEMs and sampling mask were based on different plate rotation models, could this be a source of discrepancy, particularly given how small your sampling cells are?

L118 - Why did you choose to use a 1° grid rather than equal area, which some may consider to be "fairer"?

L124 - Surely this is the cells within the shallow marine grid, rather than without?

L126 - Please describe what a summed minimum-spanning tree is, or how it is calculated.

L153 - Are differences always the same sign, or is D the sum of the modulus of d? While absolute richness can only be undersampled, surely proportional richness could be overestimated for a given bin if the others are severely undersampled? I think this is something that should be clarified.

L179 - I think you only need 'occurrences' or 'collections' rather than both.

e.g. L186 - As your study interval is so long, please give periods with stages throughout.

L188 - I would argue that 0% SSC is a complete data absence!

L193-198 - I think that if you are making comparisons between summed MSTs for individual latitude bins, they should be normalised by bin area. Surely this is a control on the possible maximum value?

L208 - Perhaps change 'is limited by' to 'has no effect on the'.

L214 - Change 'types of LBG' to 'simulated LBG types'.

L226-7 - Perhaps 'Neogene (Fig. 4), making it possible to distinguish between simulated LBG types'.

L241 - Add 'LBGs' at the end of the sentence. For subsequent sentence, change to e.g. 'The ability to recover different simulated LBG types varies across the time series.'

L251-253 - I don't understand the difference between this result and that reported in L245-247.

L262-6 - While I appreciate that it may have been done for the sake of completeness, I don't think you need to report whether richness peaks match for flat LBGs, as by definition, surely they do not have clear peaks, and therefore it doesn't really matter whether these 'peaks' match. In my opinion, reporting these values in Table S3 is sufficient for interested readers.

L267 - Is the increase in peak detectability for bimodal LDG types influenced by the fact that there are twice as many peaks, and is any allowance made for this?

L286 - Please report the global richness recovered post-sampling, either a mean with standard deviation or a range would be helpful.

L298-9 - Perhaps 'indicate that LBGs based on the fossil record are a poor representation of the true distribution of past biodiversity'.

e.g. L324-7, L366-78 - This may be a personal choice of tone, but - while I appreciate the points made throughout the discussion, I think you are perhaps being overly pessimistic. The sampling masks that you applied represent the current state of the Paleobiology Database, which means that spatial sampling can only improve in future as more literature is accounted for and more fieldwork is undertaken. While it is true that gaps will always remain, the coverage of the Paleobiology Database is strongly skewed by the personal interests of those entering the data, and I think that concerted effort to improve spatial coverage could make a difference, at least in reducing the number of completely empty spatio-temporal bins. I personally would push the potential benefits of future database improvement (e.g. making L428 stronger), rather than only

underlining the inadequacy of the fossil record for addressing these questions.

L355 - Change to e.g. 'failing to differentiate between artificially similar palaeolatitudinal peaks'.

L377-8 - I'm not sure I understand, please can you elaborate on this sentence?

L386-8 - Olson's Extinction (as examined by Brocklehurst et al. 2017) is a purported example of this, which you could mention here.

L403 - I disagree that a 1° grid is coarse, particularly for the older end of your time interval, due to uncertainty in plate rotation models so far back in time. Your spatial resolution should be mentioned here as an influential factor on your results, but I don't think you need to be quite so negative.

L412-3 - I think using the word 'significant' outside of 'statistical significance' is difficult and should be avoided. I think you could also explicitly say in the second half of this sentence that terrestrial LBGs may be influenced by different drivers (e.g. precipitation) which could in turn influence their relationship with sampling.

L430 - Add 'caution should be exercised when attempting'.

L433-46 - The conclusions are well written and an excellent summary of the important findings. All of the figures, including the supplement, are clear and informative, and well explained by their captions. The GitHub repository is also well organised and comprehensive, and I particularly like the GIF, which is a great addition to the materials provided.

Review form: Reviewer 2

Recommendation

Major revision is needed (please make suggestions in comments)

Scientific importance: Is the manuscript an original and important contribution to its field?

Excellent

General interest: Is the paper of sufficient general interest?

Excellent

Quality of the paper: Is the overall quality of the paper suitable?

Acceptable

Is the length of the paper justified?

Yes

Should the paper be seen by a specialist statistical reviewer?

No

Do you have any concerns about statistical analyses in this paper? If so, please specify them explicitly in your report.

Yes

It is a condition of publication that authors make their supporting data, code and materials available - either as supplementary material or hosted in an external repository. Please rate, if applicable, the supporting data on the following criteria.

Is it accessible?

Yes

Is it clear?

Yes

Is it adequate?

Yes

Do you have any ethical concerns with this paper?

No

Comments to the Author

Jones et al investigated whether the current sampling coverage of the fossil record can recover the "true" latitudinal biodiversity gradients (LBG) in simulated scenarios. For each geologic stage throughout the last ~ 300 Myr, they first simulated three types of LBG: flat, unimodal and bimodal (simulated diversity) based on the frequency distribution of range sizes in the modern marine biota. They then used the spatial-temporal information of the fossil occurrences in the Paleobiology Database (PBDB) to sample the simulated biodiversity patterns (sampled diversity). They also used the classic rarefaction approach to control for potential sampling effects in sampled diversity (standardized diversity). To assess the effectiveness of sampling, they assessed a) the similarity between simulated, sampled and standardized diversity across latitudinal bins for each stage, and b) the similarity in LBG patterns from the three LBG scenarios generated by the simulation, and later sampled and standardized data. Based on their results, they concluded that the current sampling coverage of the fossil record makes recovering true LBG challenging, even after controlling for sampling variation using rarefaction.

I think this is a very important exploration of a critical topic in biodiversity research -- what biological questions can we (not) answer with the current data. The potential issues of sampling variation, due to the rarity of samples and/or arbitrary variation in sampling efforts, deserve more highlight than currently given in many fields, paleo or neontological biology. Therefore, I think this paper will be of great interest for the broad readership of the journal and will make a impact. I enjoyed reading many parts of the paper very much and really appreciate the authors' effort to push this topic to the spotlight. I hope my comments below could help improve the clarity of the paper to make it a nice publication.

In particular, I see the most important point of the paper being the challenge of recovering underlying "true" LBG scenario, mostly due to the biased sampling to mid-latitudes in the Northern Hemisphere. This point could be made standing out more with some simplification of other contents. Like I said, I enjoyed reading a lot of the discussion of the topic, particularly the introduction. I also think the analyses are comprehensive for a single paper, but the description of the methodology and results could use some revision. There are too many details in the current version, and not very well organized, so reading these sections was very difficult for me. One easy way out I could suggest would be to move most of the information about the sampled diversity to the supplementary and focus on comparing the simulated and standardized diversity in the main text. At least with regard to deep-time LBG, I hope we can all agree that we cannot trust the sampled diversity without any control for sampling. The result that even standardized diversity does not track the simulated patterns very well is a way more crucial point to make for the field. With that said, I also wonder if rarefaction should be presented as a representative of all the methods out there (especially when it doesn't work well). At the very least, I would hope to see at least one extrapolation approach like Chao2 or the similar.

I also don't see the point of the summed minimum-spanning tree (MST). Based on the not so clear description, I assume the MST is based on the geographic coordinates of the samples? Does this roughly measure the spatial extension of the sampled areas? In the case of studying LBG, maybe the latitudinal span is more important than the longitudinal span? In any case, clarity is needed for explaining this analysis (especially the purpose of it).

The authors also emphasized the incompleteness of the sampling, both in the results (e.g. L179-189) and in the discussion (e.g. L298-299). The presented data are based on the simulated diversity, which might as well be too low compared to reality so that the real coverage could be even more extreme, but none of these is really crucial for recovering the underlying LBG scenario

if the sampling had not been so biased and thus inefficient for recovering at least some LBG patterns. Therefore, I think these low sampling covering needs to be put into context, or else quite meaningless. I think people in the field know that we can never sample the true diversity, past or present, but that's not what might stop us from asking questions -- it's the biases in sampling that cause the major problems and, how such biases might distort the biological patterns is the main point here, in my opinion.

Another point that gets discussed repeatedly is the lack of sampling in some regions/time bins. Again, I think these are not so relevant here because typical analyses would not treat them as true absence (as suggested by L362-363) but rather, just focus on the time bins (or regions) with more coverage in the database. As currently written, I cannot see why these absence of data would be a big problem -- not that I don't think they are problematic, but the relevance is not explained clearly here (e.g. L260-261; 359-363). The suggestion of using niche/occupancy modelling for such absence thus seems a bit out of nowhere (L424-426) -- a bit more explanation is needed, I think.

Some minor points:

L124: did you mean cells "with" shallow marine grid?

L153-156: very confusing description about absolute difference between proportional richness curves. Proportional to what?

L277-278: not sure why the simulated richness would be expected to correlate with sampling metrics

L287-288: again, not sure how these could be relevant to extinction/radiation analyses, especially most people would not consider absolute loss/gain of diversity when interpreting fossil data, I hope.

Decision letter (RSPB-2020-2762.R0)

22-Dec-2020

Dear Dr Jones:

Your manuscript has now been peer reviewed and the reviews have been assessed by an Associate Editor. The reviewers' comments (not including confidential comments to the Editor) and the comments from the Associate Editor are included at the end of this email for your reference. As you will see, the reviewers and the Editors have raised some concerns with your manuscript and we would like to invite you to revise your manuscript to address them.

Research ethics:

Use of animals and field studies:

It is a condition of publication that you make available the data and research materials supporting the results in the article. Please see our Data Sharing Policies (<https://royalsociety.org/journals/authors/author-guidelines/#data>). Datasets should be deposited in an appropriate publicly available repository and details of the associated accession number, link or DOI to the datasets must be included in the Data Accessibility section of the article (<https://royalsociety.org/journals/ethics-policies/data-sharing-mining/>). Reference(s) to datasets should also be included in the reference list of the article with DOIs (where available).

Please submit a copy of your revised paper within three weeks. If we do not hear from you within this time your manuscript will be rejected. If you are unable to meet this deadline please let us know as soon as possible, as we may be able to grant a short extension.

Best wishes,
Dr Daniel Costa
mailto:proceedingsb@royalsociety.org

Associate Editor
Board Member: 1
Comments to Author:

Both reviewers were positive but had a number of comments that should substantially improve the manuscript. The comments in Rev 1's review have some very important issues that need addressing. Reviewer 2 in particular had some explicit suggestions about organization of text and SOM; the addition of extrapolation approaches, such as Chao2 or similar; further justification/explanation for using MSTs; and putting low sampling in the proper context. Please address all of the reviewer's concerns in your revision.

Reviewer(s)' Comments to Author:

Referee: 1

Comments to the Author(s)

The manuscript by Jones et al., on the extent to which sampling bias influences our perception of spatial biodiversity patterns in deep time, is an excellent contribution. The analyses are well considered and thorough, and the manuscript is clearly written and accessible given the complexity of the methodology. I consider it to fit well within the remit of Proceedings B, and suggest only minor changes to the manuscript, as outlined below, prior to publication.

L20 - Maybe replace 'indistinguishable' with 'obscured'.

L26 - I would replace 'obscure' here with 'prevent' or 'inhibit'.

L32 - Might be worth adding the qualifier 'future climate change', or similar ('anthropogenic?').

L37 - The first question here is somewhat unnecessary.

L39 - I'm unsure about the word 'grain', perhaps 'scale' and/or 'resolution' (unless you meant something else?) would be more accurate.

L41 - This is pedantic (!) but 'decreases' implies a directional or linear variable, which 'regions' aren't strictly, so despite a possible reluctance to repeat 'latitudes' I think that would be better, or another similar rewording.

L43 - Replace 'it', perhaps 'large-scale spatial variation in biodiversity'.

e.g. L53 - I think you should cite Antell et al. (doi.org/10.1016/j.cub.2019.10.065) as another example considering spatial sampling bias in the marine fossil record.

L58 - 'taxon pool' feels ambiguous between a pool of occurrences and the taxa counted within it. You could be explicit about both, i.e. variation in geographic area -> variation in sample size of occurrences -> strong control on variation in taxon counts.

L62 - You could remove 'in data originating'.

L63 - Change to e.g. '(flat, unimodal and bimodal) using palaeogeographies of 56'.

L64 - Why did you use this time interval?

- L68-9 - I find this sentence confusing. Both simulated and sampled curves were produced based on the DEMs, but surely only the sampled curve was subjected to the sampling window filter? The second half of the sentence then seems to contradict your use of 'known' in L74.
- L78 - It would be more accurate to specify 'life' as 'marine invertebrate' here.
- L95 - Could you give a range (or average) of range sizes?
- L98 - I think 'We did so...' is an unnecessary sentence.
- L99-102 - I think this interpretation could be explained in a way which is more understandable, for example 'Our simulated species had range sizes and frequency distributions which replicated those of sponges and echinoderms, but were less similar to those of bivalves and anthozoans' (if I've understood your table correctly?). And/or you could change the sentiment in L101-2 to something along the lines of 'Empirical data from these clades show that they vary in their range sizes and occurrence frequency distributions, and our intention was to simulate a mixed community of these animals rather than to precisely replicate any one of them, therefore the simulation performs well'. You could probably write this more succinctly than I have! Otherwise, change 'both' to 'a mixture of', or similar, and add 'simulated and empirical groups, and the metric used to compare them'.
- L101 - Perhaps a point to add to the discussion, if not here: do you think differences in the amount of time averaging between your fossil data and OBIS observations could make them an unfair comparison for range sizes and frequency distributions?
- L103-111 - I think this paragraph should be inserted into the one above, around L88, to save coming back to how the probability grids were used.
- L110 - Change 'between' to 'in those'.
- L114 - Add 'real fossil collections'.
- L116 - Was this "all marine", or literally everything? If everything, could the inclusion of terrestrial collections in coastal grid cells have influenced your sampling masks?
- L117-120 - Were these the palaeo-coordinates produced by the PBDB's model, i.e. the GPlates model? If your DEMs and sampling mask were based on different plate rotation models, could this be a source of discrepancy, particularly given how small your sampling cells are?
- L118 - Why did you choose to use a 1° grid rather than equal area, which some may consider to be "fairer"?
- L124 - Surely this is the cells within the shallow marine grid, rather than without?
- L126 - Please describe what a summed minimum-spanning tree is, or how it is calculated.
- L153 - Are differences always the same sign, or is D the sum of the modulus of d? While absolute richness can only be undersampled, surely proportional richness could be overestimated for a given bin if the others are severely undersampled? I think this is something that should be clarified.
- L179 - I think you only need 'occurrences' or 'collections' rather than both.
- e.g. L186 - As your study interval is so long, please give periods with stages throughout.
- L188 - I would argue that 0% SSC is a complete data absence!
- L193-198 - I think that if you are making comparisons between summed MSTs for individual latitude bins, they should be normalised by bin area. Surely this is a control on the possible maximum value?
- L208 - Perhaps change 'is limited by' to 'has no effect on the'.
- L214 - Change 'types of LBG' to 'simulated LBG types'.
- L226-7 - Perhaps 'Neogene (Fig. 4), making it possible to distinguish between simulated LBG types'.
- L241 - Add 'LBGs' at the end of the sentence. For subsequent sentence, change to e.g. 'The ability to recover different simulated LBG types varies across the time series.'
- L251-253 - I don't understand the difference between this result and that reported in L245-247.
- L262-6 - While I appreciate that it may have been done for the sake of completeness, I don't think you need to report whether richness peaks match for flat LBGs, as by definition, surely they do not have clear peaks, and therefore it doesn't really matter whether these 'peaks' match. In my opinion, reporting these values in Table S3 is sufficient for interested readers.
- L267 - Is the increase in peak detectability for bimodal LDG types influenced by the fact that there are twice as many peaks, and is any allowance made for this?

L286 – Please report the global richness recovered post-sampling, either a mean with standard deviation or a range would be helpful.

L298-9 – Perhaps ‘indicate that LBGs based on the fossil record are a poor representation of the true distribution of past biodiversity’.

e.g. L324-7, L366-78 – This may be a personal choice of tone, but – while I appreciate the points made throughout the discussion, I think you are perhaps being overly pessimistic. The sampling masks that you applied represent the current state of the Paleobiology Database, which means that spatial sampling can only improve in future as more literature is accounted for and more fieldwork is undertaken. While it is true that gaps will always remain, the coverage of the Paleobiology Database is strongly skewed by the personal interests of those entering the data, and I think that concerted effort to improve spatial coverage could make a difference, at least in reducing the number of completely empty spatio-temporal bins. I personally would push the potential benefits of future database improvement (e.g. making L428 stronger), rather than only underlining the inadequacy of the fossil record for addressing these questions.

L355 – Change to e.g. ‘failing to differentiate between artificially similar palaeolatitudinal peaks’.

L377-8 – I’m not sure I understand, please can you elaborate on this sentence?

L386-8 – Olson’s Extinction (as examined by Brocklehurst et al. 2017) is a purported example of this, which you could mention here.

L403 – I disagree that a 1° grid is coarse, particularly for the older end of your time interval, due to uncertainty in plate rotation models so far back in time. Your spatial resolution should be mentioned here as an influential factor on your results, but I don’t think you need to be quite so negative.

L412-3 – I think using the word ‘significant’ outside of ‘statistical significance’ is difficult and should be avoided. I think you could also explicitly say in the second half of this sentence that terrestrial LBGs may be influenced by different drivers (e.g. precipitation) which could in turn influence their relationship with sampling.

L430 – Add ‘caution should be exercised when attempting’.

L433-46 – The conclusions are well written and an excellent summary of the important findings. All of the figures, including the supplement, are clear and informative, and well explained by their captions. The GitHub repository is also well organised and comprehensive, and I particularly like the GIF, which is a great addition to the materials provided.

Referee: 2

Comments to the Author(s)

Jones et al investigated whether the current sampling coverage of the fossil record can recover the "true" latitudinal biodiversity gradients (LBG) in simulated scenarios. For each geologic stage throughout the last ~ 300 Myr, they first simulated three types of LBG: flat, unimodal and bimodal (simulated diversity) based on the frequency distribution of range sizes in the modern marine biota. They then used the spatial-temporal information of the fossil occurrences in the Paleobiology Database (PBDB) to sample the simulated biodiversity patterns (sampled diversity). They also used the classic rarefaction approach to control for potential sampling effects in sampled diversity (standardized diversity). To assess the effectiveness of sampling, they assessed a) the similarity between simulated, sampled and standardized diversity across latitudinal bins for each stage, and b) the similarity in LBG patterns from the three LBG scenarios generated by the simulation, and later sampled and standardized data. Based on their results, they concluded that the current sampling coverage of the fossil record makes recovering true LBG challenging, even after controlling for sampling variation using rarefaction.

I think this is a very important exploration of a critical topic in biodiversity research -- what biological questions can we (not) answer with the current data. The potential issues of sampling variation, due to the rarity of samples and/or arbitrary variation in sampling efforts, deserve more highlight than currently given in many fields, paleo or neontological biology. Therefore, I think this paper will be of great interest for the broad readership of the journal and will make a impact. I enjoyed reading many parts of the paper very much and really appreciate the authors'

effort to push this topic to the spotlight. I hope my comments below could help improve the clarity of the paper to make it a nice publication.

In particular, I see the most important point of the paper being the challenge of recovering underlying "true" LBG scenario, mostly due to the biased sampling to mid-latitudes in the Northern Hemisphere. This point could be made standing out more with some simplification of other contents. Like I said, I enjoyed reading a lot of the discussion of the topic, particularly the introduction. I also think the analyses are comprehensive for a single paper, but the description of the methodology and results could use some revision. There are too many details in the current version, and not very well organized, so reading these sections was very difficult for me.

One easy way out I could suggest would be to move most of the information about the sampled diversity to the supplementary and focus on comparing the simulated and standardized diversity in the main text. At least with regard to deep-time LBG, I hope we can all agree that we cannot trust the sampled diversity without any control for sampling. The result that even standardized diversity does not track the simulated patterns very well is a way more crucial point to make for the field. With that said, I also wonder if rarefaction should be presented as a representative of all the methods out there (especially when it doesn't work well). At the very least, I would hope to see at least one extrapolation approach like Chao2 or the similar.

I also don't see the point of the summed minimum-spanning tree (MST). Based on the not so clear description, I assume the MST is based on the geographic coordinates of the samples? Does this roughly measure the spatial extension of the sampled areas? In the case of studying LBG, maybe the latitudinal span is more important than the longitudinal span? In any case, clarity is needed for explaining this analysis (especially the purpose of it).

The authors also emphasized the incompleteness of the sampling, both in the results (e.g. L179-189) and in the discussion (e.g. L298-299). The presented data are based on the simulated diversity, which might as well be too low compared to reality so that the real coverage could be even more extreme, but none of these is really crucial for recovering the underlying LBG scenario if the sampling had not been so biased and thus inefficient for recovering at least some LBG patterns. Therefore, I think these low sampling covering needs to be put into context, or else quite meaningless. I think people in the field know that we can never sample the true diversity, past or present, but that's not what might stop us from asking questions -- it's the biases in sampling that cause the major problems and, how such biases might distort the biological patterns is the main point here, in my opinion.

Another point that gets discussed repeatedly is the lack of sampling in some regions/time bins. Again, I think these are not so relevant here because typical analyses would not treat them as true absence (as suggested by L362-363) but rather, just focus on the time bins (or regions) with more coverage in the database. As currently written, I cannot see why these absence of data would be a big problem -- not that I don't think they are problematic, but the relevance is not explained clearly here (e.g. L260-261; 359-363). The suggestion of using niche/occupancy modelling for such absence thus seems a bit out of nowhere (L424-426) -- a bit more explanation is needed, I think.

Some minor points:

L124: did you mean cells "with" shallow marine grid?

L153-156: very confusing description about absolute difference between proportional richness curves. Proportional to what?

L277-278: not sure why the simulated richness would be expected to correlate with sampling metrics

L287-288: again, not sure how these could be relevant to extinction/radiation analyses, especially most people would not consider absolute loss/gain of diversity when interpreting fossil data, I hope.

Author's Response to Decision Letter for (RSPB-2020-2762.R0)

See Appendix A.

Decision letter (RSPB-2020-2762.R1)

28-Jan-2021

Dear Dr Jones

I am pleased to inform you that your manuscript entitled "Spatial sampling heterogeneity limits the detectability of deep time latitudinal biodiversity gradients" has been accepted for publication in Proceedings B.

Open Access

You are invited to opt for Open Access, making your freely available to all as soon as it is ready for publication under a CC BY licence. Our article processing charge for Open Access is £1700. Corresponding authors from member institutions (<http://royalsocietypublishing.org/site/librarians/allmembers.xhtml>) receive a 25% discount to these charges. For more information please visit <http://royalsocietypublishing.org/open-access>.

Your article has been estimated as being 9 pages long. Our Production Office will be able to confirm the exact length at proof stage.

Paper charges

Sincerely,
Dr Daniel Costa
Editor, Proceedings B
mailto: proceedingsb@royalsociety.org

Associate Editor:
Comments to Author:
Congratulations on revising your paper with careful attention to the reviewer comments. I look forward to seeing the paper in print.

Appendix A

18 January 2021

Dear Editor and Reviewers,

We would first like to extend our gratitude for handling and reviewing our manuscript, particularly with the current COVID-19 pandemic. Below, we respond to each individual point raised. Our modifications to the original manuscript are attached in a tracked format, and a 'clean' version is also submitted. Comments from the referees here are in *italics*, while our responses are in **bold**. We note that we have implemented the vast majority of the referees' comments, and highlight that our conclusions remain unchanged, despite these modifications. A small number of comments made by the referees have not been implemented. Principally, this is due to being beyond the scope of this work, and would only further complicate an already technical study. Detailed responses to each comment are provided below.

Yours sincerely,

Dr Lewis A. Jones

Referee: 1

The manuscript by Jones et al., on the extent to which sampling bias influences our perception of spatial biodiversity patterns in deep time, is an excellent contribution. The analyses are well considered and thorough, and the manuscript is clearly written and accessible given the complexity of the methodology. I consider it to fit well within the remit of Proceedings B, and suggest only minor changes to the manuscript, as outlined below, prior to publication.

We thank the reviewer for taking the time to review our manuscript and their positive comments.

L20 – Maybe replace 'indistinguishable' with 'obscured'.

Thank you, we have implemented this change.

L26 – I would replace 'obscure' here with 'prevent' or 'inhibit'.

Thank you, we have updated obscure to inhibit, as suggested.

L32 – Might be worth adding the qualifier 'future climate change', or similar ('anthropogenic?').

Thank you, we have amended this to 'future climate change'.

L37 – The first question here is somewhat unnecessary.

We have opted to retain this as we feel it sets context.

L39 – I'm unsure about the word 'grain', perhaps 'scale' and/or 'resolution' (unless you meant something else?) would be more accurate.

We have updated 'grain' to 'scale', as recommended by the reviewer.

L41 – This is pedantic (!) but 'decreases' implies a directional or linear variable, which 'regions' aren't strictly, so despite a possible reluctance to repeat 'latitudes' I think that would be better, or another similar rewording.

This is a fair point. We have updated 'regions' to 'latitudes'. Thank you.

L43 – Replace ‘it’, perhaps ‘large-scale spatial variation in biodiversity’.

Thank you, we have updated accordingly.

e.g. L53 – I think you should cite Antell et al. (doi.org/10.1016/j.cub.2019.10.065) as another example considering spatial sampling bias in the marine fossil record.

Thank you. This is now included.

L58 – ‘taxon pool’ feels ambiguous between a pool of occurrences and the taxa counted within it. You could be explicit about both, i.e. variation in geographic area -> variation in sample size of occurrences -> strong control on variation in taxon counts.

This is an excellent point; we have updated accordingly.

L62 – You could remove ‘in data originating’.

Thank you, we have now updated this accordingly.

L63 – Change to e.g. ‘(flat, unimodal and bimodal) using palaeogeographies of 56’.

Updated accordingly. Thank you.

L64 – Why did you use this time interval?

We have now added a sentence here with justification: “This time period covers a range of major Earth system changes, including transitions from greenhouse to icehouse climatic regimes, which are considered to be a major driver in the spatial distribution of biodiversity”.

L68-9 – I find this sentence confusing. Both simulated and sampled curves were produced based on the DEMs, but surely only the sampled curve was subjected to the sampling window filter? The second half of the sentence then seems to contradict your use of ‘known’ in L74.

We agree that this was confusing. We have revised this sentence to: “Simulated (using the raw dataset) and sampled (using the filtered dataset) palaeolatitudinal biodiversity curves were then constructed, and compared to quantify the data lost between the ‘known’ and sampled fossil record.”

L78 – It would be more accurate to specify ‘life’ as ‘marine invertebrate’ here.

Updated accordingly. Thank you.

L95 – Could you give a range (or average) of range sizes?

We have now added a summary statistics table in the Supplementary Material (Table S1) with range sizes. We have refrained from adding this into the main text due to space restrictions.

L98 – I think ‘We did so...’ is an unnecessary sentence.

We have now removed this sentence.

L99-102 – I think this interpretation could be explained in a way which is more understandable, for example ‘Our simulated species had range sizes and frequency distributions which replicated those of sponges and echinoderms, but were less similar to those of bivalves and anthozoans’ (if I’ve understood your table correctly?). And/or you could change the sentiment in L101-2 to something along the lines of ‘Empirical data from these clades show that they vary in their range sizes and occurrence frequency distributions, and our intention was to simulate a mixed community of these animals rather than to precisely replicate any one of them, therefore the simulation performs well’. You could probably write

this more succinctly than I have! Otherwise, change ‘both’ to ‘a mixture of’, or similar, and add ‘simulated and empirical groups, and the metric used to compare them’.

Thank you, we have now updated this accordingly.

L101 – Perhaps a point to add to the discussion, if not here: do you think differences in the amount of time averaging between your fossil data and OBIS observations could make them an unfair comparison for range sizes and frequency distributions?

We agree that time averaging effects in the fossil record likely influence observed range sizes and frequency distributions. However, as the fossil record is inherently incomplete, using fossil data to estimate range sizes and frequency distributions would only bias our simulated species as this data is already exposed to a number of filters (e.g. sampling bias). Therefore, modern occurrence data is expected to provide a better approximation of species distributions in life, before applying any sampling/filtering window. We have now added a couple of sentences in the discussion highlighting this possible limitation.

L103-111 – I think this paragraph should be inserted into the one above, around L88, to save coming back to how the probability grids were used.

Thank you, we have now updated this accordingly.

L110 – Change ‘between’ to ‘in those’.

Thank you, we have now updated this accordingly.

L114 – Add ‘real fossil collections’.

Thank you, we have now updated this accordingly.

L116 – Was this “all marine”, or literally everything? If everything, could the inclusion of terrestrial collections in coastal grid cells have influenced your sampling masks?

We originally used all collections, with the intention of catching these coastal grid cells. As species could be simulated within these cells, and ‘coastal’ grid cells should represent the transition between ocean and land, we felt it was valid to include these collections. However, we have now restricted our use of collections to just marine collections to address the reviewer’s concerns. We have rerun our entire analyses using this more restricted dataset – although it has had a small effect on our results, the overall trends and conclusions are the same.

L117-120 – Were these the palaeo-coordinates produced by the PBDB’s model, i.e. the GPlates model? If your DEMs and sampling mask were based on different plate rotation models, could this be a source of discrepancy, particularly given how small your sampling cells are?

We used the standard PBDB palaeocoordinates, as high-resolution palaeorotations from the Getech Plate model were not available to us, and we were limited to a spatially aggregated model with a resolution of 3.75° x 2.5°. We tested the applicability of these low-resolution palaeorotations. However, due to their coarse resolution, they were not suitable for producing our 1° x 1° spatial sampling grids. As such, we have opted to retain the GPlates rotations, and have added a section about this in our discussion.

L118 – Why did you choose to use a 1° grid rather than equal area, which some may consider to be “fairer”?

Degree grids are used as standard in simulation studies, including in palaeoclimatology, macroecology and palaeobiology [1–5]. For our simulations we followed suit, opting to use a 1° x

1° grid. Principally, this was to enable faster computation times and to prevent the unnecessary transformation of Earth System model and occurrence data.

L124 – Surely this is the cells within the shallow marine grid, rather than without?

Yes, this was a mistake. We have updated accordingly. Thank you for highlighting this.

L126 – Please describe what a summed minimum-spanning tree is, or how it is calculated.

Thank you, we have now updated this accordingly.

L153 – Are differences always the same sign, or is D the sum of the modulus of d ? While absolute richness can only be undersampled, surely proportional richness could be overestimated for a given bin if the others are severely undersampled? I think this is something that should be clarified.

In our study, d represents the absolute (i.e. non-negative value) difference in richness between curves for each palaeolatitudinal bin for $j = 1, 2, \dots, n$ palaeolatitudinal bins, and D is the sum of this difference across palaeolatitudinal bins. We have now updated this section to make it clearer that we refer to the absolute value. We agree with the reviewer that proportional richness can be over or underestimated, and this is in fact the point of this metric. When studying the latitudinal biodiversity gradient, we are interested in the shape of the diversity curve/steepness of the gradient, not the true richness values. This is described in our methods paragraph: ‘Latitudinal biodiversity gradient analyses’.

L179 – I think you only need ‘occurrences’ or ‘collections’ rather than both.

We have removed occurrences. Thank you.

e.g. L186 – As your study interval is so long, please give periods with stages throughout.

Thank you, we agree that this is appropriate and have updated the manuscript accordingly.

L188 – I would argue that 0% SSC is a complete data absence!

We agree! We have updated this sentence to read how we originally intended: “Furthermore, ~40% of palaeolatitudinal bins have a SSC of 0%, indicating substantial data absence along our time series.”

L193-198 – I think that if you are making comparisons between summed MSTs for individual latitude bins, they should be normalised by bin area. Surely this is a control on the possible maximum value?

This is an interesting point raised by the reviewer and is something we previously considered. However, we decided against implementing it as in many ways it is not the point we are trying to address. In our usage, we quantify spatial sampling extent using summed MST to investigate its impact on sampled diversity. Although maximum possible spatial sampling extent may vary between latitudinal bins due to available area, this does necessitate the normalisation of this data (in such a case, diversity should also be normalised due to the species-area effect) as we are interested in how this drives observed diversity patterns. In addition, normalising by area would not have the desired effect, as it is also the distribution of this area which is important when calculating the summed MST (e.g. X number of cells clustered together, in our shallow marine grids, would have a lower summed MST than the same number of cells highly dispersed). Nevertheless, our spatial sampling coverage metric does provide a normalised perspective on the amount of spatial sampling relative to spatial sampling opportunities.

L208 – Perhaps change ‘is limited by’ to ‘has no effect on the’.

Thank you, we have now updated this accordingly.

L214 – Change ‘types of LBG’ to ‘simulated LBG types’.

Thank you, we have now updated this accordingly.

L226-7 – Perhaps ‘Neogene (Fig. 4), making it possible to distinguish between simulated LBG types’.

Thank you, we have now updated this accordingly.

L241 – Add ‘LBGs’ at the end of the sentence. For subsequent sentence, change to e.g. ‘The ability to recover different simulated LBG types varies across the time series.’

Thank you, we have now updated this accordingly.

L251-253 – I don’t understand the difference between this result and that reported in L245-247.

This line refers to the statistical significance of Pearson’s r scores. L245–247 refers to the results from testing for difference between the Pearson’s r scores for pair-wise combinations between simulated, sampled and sampling-standardised LBGs.

L262-6 – While I appreciate that it may have been done for the sake of completeness, I don’t think you need to report whether richness peaks match for flat LBGs, as by definition, surely they do not have clear peaks, and therefore it doesn’t really matter whether these ‘peaks’ match. In my opinion, reporting these values in Table S3 is sufficient for interested readers.

We appreciate the reviewer’s comment and had considered this modification ourselves. However, we feel that it is worth including as we still observe substantial differences between simulated flat LBGs and their sampled counterparts. As it has been suggested that there were flat-type LBGs in the geological past [e.g. 6,7], we feel it warrants its inclusion. In addition, in a previous review of this work, we only focused on unimodal and bimodal LBGs. However, it was suggested by several reviewers to include null/flat LBGs in our analyses for the sake of completeness.

L267 – Is the increase in peak detectability for bimodal LDG types influenced by the fact that there are twice as many peaks, and is any allowance made for this?

Although our simulated bimodal LBGs do have two peaks in richness, one palaeolatitudinal bin always has a higher richness in our analyses due to the heterogeneous distribution of shallow marine area (maximum peak). Despite this, as we calculate both the exact palaeolatitudinal bin with peak richness, as well as the correct palaeolatitudinal region (tropics, temperate or polar), we already capture this variability. It is possible that having two peaks in a bimodal LBG may influence the detectability of the peak in the right palaeolatitudinal zone, but it would not do so for the exact palaeolatitudinal bin.

L286 – Please report the global richness recovered post-sampling, either a mean with standard deviation or a range would be helpful.

We have now added the mean and standard deviation of sampled richness. Thank you.

L298-9 – Perhaps ‘indicate that LBGs based on the fossil record are a poor representation of the true distribution of past biodiversity’.

Thank you, we have now updated this accordingly.

e.g. L324-7, L366-78 – This may be a personal choice of tone, but – while I appreciate the points made throughout the discussion, I think you are perhaps being overly pessimistic. The sampling masks that you applied represent the current state of the Paleobiology Database, which means that spatial sampling can only improve in future as more literature is accounted for and more fieldwork is undertaken. While it is true that gaps will always remain, the coverage of the Paleobiology Database

is strongly skewed by the personal interests of those entering the data, and I think that concerted effort to improve spatial coverage could make a difference, at least in reducing the number of completely empty spatio-temporal bins. I personally would push the potential benefits of future database improvement (e.g. making L428 stronger), rather than only underlining the inadequacy of the fossil record for addressing these questions.

Thank you, we have now updated the manuscript to be less pessimistic, but still highlight the issues with the current data. We agree that with data collection in under-sampled regions and further improvements to databases (i.e. the Paleobiology Database), we can still hope to address these questions in the future. We have now made this clear in the discussion of our manuscript. However, we would like to highlight that even with improved sampling, the fossil record is biased by the heterogenous distribution of available outcrop, which is an issue that even fair sampling will not be able to overcome due to data absence.

L355 – Change to e.g. ‘failing to differentiate between artificially similar palaeolatitudinal peaks’.

Thank you. We have updated accordingly.

L377-8 – I’m not sure I understand, please can you elaborate on this sentence?

We updated this sentence with an example to improve clarity: “Nevertheless, skewed palaeolatitudinal sampling might be negated at higher SSC due to latitudinal differences in species discovery curves (e.g. if diversity is greater at lower latitudes, more species may be discovered in smaller sampled areas at lower latitudes, than larger sampled areas at higher latitudes).

L386-8 – Olson’s Extinction (as examined by Brocklehurst et al. 2017) is a purported example of this, which you could mention here.

Thank you, this has now been included.

L403 – I disagree that a 1° grid is coarse, particularly for the older end of your time interval, due to uncertainty in plate rotation models so far back in time. Your spatial resolution should be mentioned here as an influential factor on your results, but I don’t think you need to be quite so negative.

We have updated this section accordingly, removing the reference to coarse resolution.

L412-3 – I think using the word ‘significant’ outside of ‘statistical significance’ is difficult and should be avoided. I think you could also explicitly say in the second half of this sentence that terrestrial LBGs may be influenced by different drivers (e.g. precipitation) which could in turn influence their relationship with sampling.

Agreed, we have updated accordingly. Thank you.

L430 – Add ‘caution should be exercised when attempting’.

Thank you, we have now updated this accordingly.

L433-46 – The conclusions are well written and an excellent summary of the important findings. All of the figures, including the supplement, are clear and informative, and well explained by their captions. The GitHub repository is also well organised and comprehensive, and I particularly like the GIF, which is a great addition to the materials provided.

Thank you, we really appreciate you saying so.

Referee: 2

Jones et al investigated whether the current sampling coverage of the fossil record can recover the "true" latitudinal biodiversity gradients (LBG) in simulated scenarios. For each geologic stage throughout the last ~ 300 Myr, they first simulated three types of LBG: flat, unimodal and bimodal (simulated diversity) based on the frequency distribution of range sizes in the modern marine biota. They then used the spatial-temporal information of the fossil occurrences in the Paleobiology Database (PBDB) to sample the simulated biodiversity patterns (sampled diversity). They also used the classic rarefaction approach to control for potential sampling effects in sampled diversity (standardized diversity). To assess the effectiveness of sampling, they assessed a) the similarity between simulated, sampled and standardized diversity across latitudinal bins for each stage, and b) the similarity in LBG patterns from the three LBG scenarios generated by the simulation, and later sampled and standardized data. Based on their results, they concluded that the current sampling coverage of the fossil record makes recovering true LBG challenging, even after controlling for sampling variation using rarefaction.

I think this is a very important exploration of a critical topic in biodiversity research -- what biological questions can we (not) answer with the current data. The potential issues of sampling variation, due to the rarity of samples and/or arbitrary variation in sampling efforts, deserve more highlight than currently given in many fields, paleo or neontological biology. Therefore, I think this paper will be of great interest for the broad readership of the journal and will make an impact. I enjoyed reading many parts of the paper very much and really appreciate the authors' effort to push this topic to the spotlight. I hope my comments below could help improve the clarity of the paper to make it a nice publication.

We would like to thank the reviewer for taking the time to review our manuscript, and providing constructive comments that have improved the work presented here. We would also like to extend our gratitude for their kind words and positivity towards our manuscript. We also strongly agree that spatial bias is a very important topic that requires additional attention in many fields.

In particular, I see the most important point of the paper being the challenge of recovering underlying "true" LBG scenario, mostly due to the biased sampling to mid-latitudes in the Northern Hemisphere. This point could be made standing out more with some simplification of other contents. Like I said, I enjoyed reading a lot of the discussion of the topic, particularly the introduction. I also think the analyses are comprehensive for a single paper, but the description of the methodology and results could use some revision. There are too many details in the current version, and not very well organized, so reading these sections was very difficult for me.

We have now updated our manuscript, particularly the methodology, to ensure easier reading and comprehension. However, due to the nature of the study, the manuscript does still retain technical details, and the much-required level of detail vital to the reader. Nevertheless, we include several figures in the supplementary material to aid reader comprehension on key aspects of our paper, including a simulation workflow diagram (Fig. S1).

One easy way out I could suggest would be to move most of the information about the sampled diversity to the supplementary and focus on comparing the simulated and standardized diversity in the main text. At least with regard to deep-time LBG, I hope we can all agree that we cannot trust the sampled diversity without any control for sampling. The result that even standardized diversity does not track the simulated patterns very well is a way more crucial point to make for the field. With that said, I also wonder if rarefaction should be presented as a representative of all the methods out there (especially when it doesn't work well). At the very least, I would hope to see at least one extrapolation approach like Chao2 or the similar.

Whilst we agree with the reviewer that uncorrected sampled diversity cannot be trusted, we include this data for added completeness to our study. By including this data, we are able to show how ineffective classical rarefaction is due to data absence by comparing sampled and sampling-standardised diversity curves. In addition, the inclusion of this data is important for quantifying just how much heterogeneous spatial sampling can distort genuine patterns of biodiversity. We agree that some subsampling methods may be more effective than others at correcting biased sampling, particularly in regard to extrapolation methods. However, the issue we mainly address in this manuscript is that subsampling approaches cannot correct for complete data absence, as is often the case in the fossil record. Even extrapolation methods in such a scenario would perform poorly. We refrain from adding additional sampling-standardisation approaches to prevent further complicating an already technical manuscript (as highlighted by the reviewer). Incorporating an estimator such as Chao 2 would necessitate removing other tests which we have carried out, as well as significantly changing our simulation structure to incorporate sampling units, and as such we feel this is beyond the scope of our current study. However, this is a topic that strongly interests us, and we plan to address this issue in a project that we are currently working on. We have highlighted in our discussion that other sampling-standardisation approaches may be more effective than classical rarefaction in correcting for uneven sampling. However, we note that such methods will still suffer from complete data absence.

I also don't see the point of the summed minimum-spanning tree (MST). Based on the not so clear description, I assume the MST is based on the geographic coordinates of the samples? Does this roughly measure the spatial extension of the sampled areas? In the case of studying LBG, maybe the latitudinal span is more important than the longitudinal span? In any case, clarity is needed for explaining this analysis (especially the purpose of it).

We apologise for this not being clear from the start. We have now added further description on summed minimum-spanning trees (MST) in the manuscript. The point of MSTs is that it measures the geographic spread of the occurrence data. If data are sourced from a wider geographic spread, one would expect to capture more variability in environments and hence species. We agree that latitudinal span could be more important at a global scale, but as our analyses is conducted within fixed latitudinal bins of 15°, the 360° longitudinal range within each of these bands could be more significant for certain latitudinal bands. In addition, the importance of latitudinal and longitudinal sampling spread likely vary in both time and space, depending upon the amount of available shallow marine habitat and climatic regimes. Therefore, measuring the geographic spread is more appropriate than focusing on either longitudinal or latitudinal span.

The authors also emphasized the incompleteness of the sampling, both in the results (e.g. L179-189) and in the discussion (e.g. L298-299). The presented data are based on the simulated diversity, which might as well be too low compared to reality so that the real coverage could be even more extreme, but none of these is really crucial for recovering the underlying LBG scenario if the sampling had not been so biased and thus inefficient for recovering at least some LBG patterns. Therefore, I think these low sampling covering needs to be put into context, or else quite meaningless. I think people in the field know that we can never sample the true diversity, past or present, but that's not what might stop us from asking questions -- it's the biases in sampling that cause the major problems and, how such biases might distort the biological patterns is the main point here, in my opinion.

We absolutely agree that one will never be able to recover true diversity, and this is why in our study we only use proportional latitudinal richness curves in our analyses, and not 'absolute diversity' curves. To make this aspect clearer we have added: "We recognise that recovering 'true' levels of biodiversity in the fossil record is an impossible challenge. The use of our proportional richness curves provides an accurate understanding of the relative latitudinal

distribution of biodiversity” in the latitudinal biodiversity gradient analyses section of our manuscript.

Another point that gets discussed repeatedly is the lack of sampling in some regions/time bins. Again, I think these are not so relevant here because typical analyses would not treat them as true absence (as suggested by L362-363) but rather, just focus on the time bins (or regions) with more coverage in the database. As currently written, I cannot see why these absence of data would be a big problem -- not that I don't think they are problematic, but the relevance is not explained clearly here (e.g. L260-261; 359-363). The suggestion of using niche/occupancy modelling for such absence thus seems a bit out of nowhere (L424-426) -- a bit more explanation is needed, I think.

We respectfully disagree. Understanding which regions suffer from sampling absence is very important to constraining whether latitudinal patterns can genuinely be reconstructed for a specific stratigraphic stage, or not. While data absence is frequently acknowledged when dealing with time series analyses (i.e. global scale studies), it tends to be ignored in spatial analyses. It is quite common in the palaeobiology literature for studies to focus on whole time series [8], or evaluate latitudinal diversity patterns without quantifying the spatial sampling coverage [7,9–11]. The use of niche/occupancy modelling provides an avenue for quantifying the amount of biological data we may be missing from certain regions. We have updated this section to emphasise that point.

Some minor points:

L124: did you mean cells "with" shallow marine grid?

Yes, this was an error on our part. Thank you for highlighting this. It has been updated accordingly.

L153-156: very confusing description about absolute difference between proportional richness curves. Proportional to what?

This alludes to the earlier statement in the methods:

“As we were principally interested in the relative shape of LBGs, we normalised all (simulated, sampled, and sampling-standardised) LBGs within their respective stages on a scale from 0 to 1 to produce proportional richness curves. This was achieved by dividing the richness of each palaeolatitudinal bin by the sum of richness across palaeolatitudinal bins for each respective stage.”

We have now added a reminder of “each palaeolatitudinal bin divided by the sum of palaeolatitudinal bins” in this section, in brackets, to remind the reader of this earlier text.

L277-278: not sure why the simulated richness would be expected to correlate with sampling metrics

In a large vs. small carbonate shelf, a null expectation would be that taxonomic richness is greater in the larger carbonate shelf (i.e. species-area relationship). Likewise, one might expect the amount of biological material preserved in the large carbonate shelf to be greater than that of a small carbonate shelf. We primarily make these comparisons between simulated richness and sampling metrics to demonstrate the de-coupled relationship between the two.

L287-288: again, not sure how these could be relevant to extinction/radiation analyses, especially most people would not consider absolute loss/gain of diversity when interpreting fossil data, I hope.

In this statement, we are not referring to the absolute loss or gain of diversity. We refer to how the distribution of both genuine biodiversity patterns and sampling regimes may influence observed biodiversity patterns. For example, if a skewed sampling window was maintained

through time, and the latitudinal biodiversity gradient switched from one type of gradient (e.g. unimodal) to another (e.g. bimodal), this could result in an observed increase/decrease in diversity at global scale. Similarly, if a unimodal-type LBG was maintained throughout time, but the sampling window shifted through time, this could also result in observed increases/decreases in diversity. Extending this further, apparent radiation and extinction events could be observed just by changing the sampling window (e.g. radiation: start sampling new regions, where different species are present or extinction: no longer sampling where former identified species were present) or the distribution of diversity (e.g. species ranges shift in response to some abiotic factor, but the sampling window stays the same).

References

1. Rangel TFLVB, Diniz-Filho JAF, Colwell RK. 2007 Species richness and evolutionary niche dynamics: a spatial pattern-oriented simulation experiment. *Am. Nat.* **170**, 602–616. (doi:10.1086/521315)
2. Qiao H, Saupe EE, Soberón J, Peterson AT, Myers CE. 2016 Impacts of Niche Breadth and Dispersal Ability on Macroevolutionary Patterns. *Am. Nat.* **188**, 149–162. (doi:10.1086/687201)
3. Valdes PJ *et al.* 2017 The BRIDGE HadCM3 family of climate models:HadCM3@Bristol v1.0. *Geoscientific Model Development* **10**, 3715–3743. (doi:10.5194/gmd-2017-16)
4. Saupe EE, Myers CE, Townsend Peterson A, Soberón J, Singarayer J, Valdes P, Qiao H. 2019 Spatio-temporal climate change contributes to latitudinal diversity gradients. *Nature Ecology & Evolution* **3**, 1419–1429. (doi:10.1038/s41559-019-0962-7)
5. Saupe EE *et al.* 2020 Extinction intensity during Ordovician and Cenozoic glaciations explained by cooling and palaeogeography. *Nat. Geosci.* **13**, 65–70. (doi:10.1038/s41561-019-0504-6)
6. Rose PJ, Fox DL, Marcot J, Badgley C. 2011 Flat latitudinal gradient in Paleocene mammal richness suggests decoupling of climate and biodiversity. *Geology* **39**, 163–166. (doi:10.1130/G31099.1)
7. Song H, Huang S, Jia E, Dai X, Wignall PB, Dunhill AM. 2020 Flat latitudinal diversity gradient caused by the Permian–Triassic mass extinction. *PNAS*, 1–6. (doi:10.1073/pnas.1918953117)
8. Powell MG. 2009 The latitudinal diversity gradient of brachiopods over the past 530 million years. *The Journal of Geology* **117**, 585–594.
9. Mannion PD, Benson RB, Upchurch P, Butler RJ, Carrano MT, Barrett PM. 2012 A temperate palaeodiversity peak in Mesozoic dinosaurs and evidence for Late Cretaceous geographical partitioning. *Global Ecology and Biogeography* **21**, 898–908.
10. Kiessling W, Simpson C, Beck B, Mewis H, Pandolfi JM. 2012 Equatorial decline of reef corals during the last Pleistocene interglacial. *PNAS* **109**, 21378–21383. (doi:10.1073/pnas.1214037110)
11. Nicholson DB, Holroyd PA, Valdes P, Barrett PM. 2016 Latitudinal diversity gradients in Mesozoic non-marine turtles. *Royal Society open science* **3**, 1–8.